# Acidic Growth Conditions Promote Epithelial-to-Mesenchymal Transition to Select More Aggressive PDAC Cell Phenotypes In Vitro

**DOI:** 10.3390/cancers15092572

**Published:** 2023-04-30

**Authors:** Madelaine Magalì Audero, Tiago Miguel Amaral Carvalho, Federico Alessandro Ruffinatti, Thorsten Loeck, Maya Yassine, Giorgia Chinigò, Antoine Folcher, Valerio Farfariello, Samuele Amadori, Chiara Vaghi, Albrecht Schwab, Stephan J. Reshkin, Rosa Angela Cardone, Natalia Prevarskaya, Alessandra Fiorio Pla

**Affiliations:** 1U1003—PHYCELL—Laboratoire de Physiologie Cellulaire, Inserm, University of Lille, Villeneuve d’Ascq, 59000 Lille, France; madelaine.audero@univ-lille.fr (M.M.A.); mhy06@mail.aub.edu (M.Y.); antoine.folcher@inserm.fr (A.F.); valerio.farfariello@inserm.fr (V.F.); 2Laboratory of Cellular and Molecular Angiogenesis, Department of Life Sciences and Systems Biology, University of Turin, 10123 Turin, Italy; federicoalessandro.ruffinatti@unito.it (F.A.R.); giorgia.chinigo@unito.it (G.C.); samuele.amadori@edu.unito.it (S.A.); chiara.vaghi@unito.it (C.V.); 3Department of Biosciences, Biotechnologies and Environment, University of Bari, 70126 Bari, Italy; tiagomac94@gmail.com (T.M.A.C.); stephanjoel.reshkin@uniba.it (S.J.R.); rosaangela.cardone@uniba.it (R.A.C.); 4Institute of Physiology II, University of Münster, 48149 Münster, Germany; loeck@uni-muenster.de (T.L.); aschwab@uni-muenster.de (A.S.)

**Keywords:** acidic tumor microenvironment, acid-selection, PDAC, cell proliferation, cell adhesion, cell migration, cell invasion, EMT

## Abstract

**Simple Summary:**

Acidosis represents a key chemical marker of the Pancreatic Ductal Adenocarcinoma (PDAC) microenvironment (TME). It induces the selection of aggressive cancer cell phenotypes and promotes its progression. Here, we describe the impact of an acidic TME on different PDAC hallmarks such as proliferation, migration, extracellular matrix digestion, invasion, and epithelial–mesenchymal transition. This was executed after establishing a model of pH_e_-selected cells that were cultured for different time periods in an acidic environment and then re-acclimated back to pH_e_ 7.4. Our findings show that the acid selection contributes to PDAC cells’ response and adaptation to the hostile acidic microenvironment, which is a requirement for the acquisition of an aggressive phenotype of PDAC cells.

**Abstract:**

Pancreatic Ductal Adenocarcinoma (PDAC) is characterized by an acidic microenvironment, which contributes to therapeutic failure. So far there is a lack of knowledge with respect to the role of the acidic microenvironment in the invasive process. This work aimed to study the phenotypic and genetic response of PDAC cells to acidic stress along the different stages of selection. To this end, we subjected the cells to short- and long-term acidic pressure and recovery to pH_e_ 7.4. This treatment aimed at mimicking PDAC edges and consequent cancer cell escape from the tumor. The impact of acidosis was assessed for cell morphology, proliferation, adhesion, migration, invasion, and epithelial–mesenchymal transition (EMT) via functional in vitro assays and RNA sequencing. Our results indicate that short acidic treatment limits growth, adhesion, invasion, and viability of PDAC cells. As the acid treatment progresses, it selects cancer cells with enhanced migration and invasion abilities induced by EMT, potentiating their metastatic potential when re-exposed to pH_e_ 7.4. The RNA-seq analysis of PANC-1 cells exposed to short-term acidosis and pH_e_-selected recovered to pH_e_ 7.4 revealed distinct transcriptome rewiring. We describe an enrichment of genes relevant to proliferation, migration, EMT, and invasion in acid-selected cells. Our work clearly demonstrates that upon acidosis stress, PDAC cells acquire more invasive cell phenotypes by promoting EMT and thus paving the way for more aggressive cell phenotypes.

## 1. Introduction

Pancreatic ductal adenocarcinoma (PDAC) is the most common neoplastic condition affecting the pancreas. It is characterized by an unsatisfactory 5-year survival rate of around 10%, far below that for other forms of solid cancer [1]. In the early stages, PDAC does not manifest itself with striking symptoms and lacks useful screening tests for evaluating asymptomatic patients. This condition often delays diagnosis, when metastases have usually/already spread throughout the body [2]. To date, there are no effective therapeutic options for PDAC. This is in part due to an immunosuppressive microenvironment and a strong desmoplastic reaction that occurs during tumor development [3]. Moreover, this type of cancer is distinguished by a characteristic chemical signature, an acidic extracellular microenvironment (˂pH_e_ 6.5), common to many other malignancies, which offers important cues for its progression and aggressiveness [4,5].

The PDAC acidic microenvironment is the result of a shift to glycolytic metabolism (the “Warburg effect”) that leads to the production and excretion of lactate and H^+^, even in the presence of oxygen [4]. The excretion of these acidic products, together with the hydration of CO_2_, the aberrant vascularization which causes a continuous hypoxic state, and the overexpression and activity of acid transporters, determine a “re-versed pH gradient” with cancer cells, in contrast to normal cells, showing an alkaline pH_i_ and acidic pH_e_. This pH gradient reversal is now considered to be an emerging “hallmark of cancers” [6]. Additionally, the acidic PDAC microenvironment also finds its roots in the physiology of the exocrine pancreas, where the intermittent exposition of pancreatic ductal cells to stroma acidification results from their bicarbonate secretion across their apical membranes into pancreatic ducts. This has been considered to promote the adaptation of acid-resistant phenotypes aiding in PDAC initiation and progression in concomitance with driving mutations [4]. 

While the reversed pH gradient represents a driving force for cancer progression, tumor acidosis may disrupt this pH balance, fostering genomic instability and imposing considerable selective pressure on cells [7,8,9,10]. The acidic microenvironment of the tumor may negatively affect cancer cell survival [11,12] and promote the selection of acid-resistant cancer subclones characterized by defense mechanisms against acidotic stress, such as autophagy [13]. 

Moreover, an acidic pH_e_ in the tumor microenvironment represents an escape strategy for tumor cells to promote EMT, cell migration, and local invasion [5,13,14,15,16,17,18,19,20,21,22], posing a threat to the efficacy of PDAC therapy. The role of the acidic pH_e_ at the peritumoral regions has been previously described by the Gillies group [17,21]. It plays a role in the degradation of the extracellular matrix at the periphery of the tumor and subsequent local invasion in melanoma and breast cancers [21]. In this complex tumor microenvironment, cells adapted to acidic pH_e_ survive due to adaptive mechanisms. 

While several teams have focused on the effects of short-term extracellular acidification on different cancer cell types [9,14,23,24,25,26,27,28,29,30,31,32,33,34,35,36,37,38], these have provided a limited understanding of how tumor cells respond to this acute acidic exposure. Moreover, as regarding the effects of long-term adaptation and selection to extracellular acidosis [13,15,16,17,39,40,41,42], few data are available on PDAC [43,44] and a clear role of acid selection in cell invasion still needs to be elucidated.

Due to the lack of a comprehensive characterization of the different cell phenotypes induced by acid selection in PDAC and, in particular, re-acclimation in physiological pH in pancreatic cancer, we aimed to describe the impact of the tumor acidic microenvironment on PDAC progression. We therefore established and characterized the different phenotypic alterations occurring in two PDAC cell lines (PANC-1 and Mia PaCa-2). Cells were subjected to a short-term (4 days) exposition to acidic pH_e_ (pH_e_ 6.6), in order to mimic the early stages of pH_e_ selection and also a long-term (1 month) exposition, as the acidification of PDAC tumor microenvironment occurs over a sustained period. To evaluate the impact of acid selection on promoting local invasion, we also established a model of pH_e_-selected cells recovered in pH_e_ 7.4. Our in vitro results, confirmed by RNA-seq, indicate that low pH_e_ acts as a stressor factor during the acute exposition, limiting PDAC cell growth and viability and impairing cell adhesion and invasion. This paves the way for a more aggressive cancer cell subpopulation, induced by the epithelial–mesenchymal transition. Migratory and invasive activity rise as the acid treatment progresses. Importantly, this process further enhances metastatic potential when cells are re-exposed to pH_e_ 7.4 as can occur during local cyclic waves of acidification and alkalization. These data emphasize the importance of an extracellular acidification in cancer cell selection and adaptation to hostile environments that promotes the development of more aggressive PDAC cell phenotypes.

## 2. Materials and Methods

### 2.1. Cell Culture and Generation of PANC-1 and Mia PaCa-2 Acidic Phenotypes

Human pancreatic ductal adenocarcinoma PANC-1 cells were obtained from the Institute for Experimental Cancer Research, Christian-Albrecht-University (CAU) of Kiel, Germany. Mia PaCa-2 cells were purchased from the American Type Culture Collection (ATCC). Control cells were cultured as monolayers in RPMI 1640 growth medium (Gibco, Cat# 31870-025) supplemented with 10% FBS (Biochrom (Cambridge, UK); Cat# S0615), 2 mM L-Glutamine (Gibco (Carlsbad, CA, USA); Cat# 25030-024), 1 mM sodium pyruvate (Gibco; Cat# 11360039), and antibiotics (penicillin/streptomycin 100 U mL^−1^; Life Technologies, Inc. (Carlsbad, CA, USA); Cat# 15070-063). For pH adjustments of the cell culture media to pH_e_ 6.6, RPMI 1640 powder medium (Sigma (Macquarie Park, NSW, Australia); Cat# R6504) was complemented with NaHCO_3_, according to the Henderson–Hasselbalch equation to derive the target pH_e_. The osmolarity of the medium was balanced using NaCl. The different powder components were dissolved in UltraPure distilled water (Invitrogen (Waltham, MA, USA); Cat# 10977-035), and the resulting medium was filtered in a sterile environment and supplemented with 10% FBS (Biochrom; Cat# S0615), 1 mM sodium pyruvate (Gibco; Cat# 11360039), and antibiotics (penicillin/streptomycin 100 U mL^−1^; Life Technologies, Inc.; Cat# 15070-063). All cells were maintained at 37 °C in a humidified 5% CO_2_ atmosphere. 

For the generation of the acidic phenotypes, PANC-1 and Mia PaCa-2 cells were exposed to pH_e_ 6.6 for different time periods by changing the media pH_e_ from 7.4 to 6.6. Control cells were cultured at pH_e_ 7.4. For studying early stages of low pH_e_- selection («4 days pH_e_ 6.6» cells), freshly split PANC-1 and Mia PaCa-2 cells were cultured in pH_e_ 6.6 medium for 4 days. PDAC mimicking cells («pH_e_-selected» cells) were generated by a 1 month-long exposure to the acidic medium before performing the experiments. pH_e_-selected + 7.4 cells were maintained in pH_e_ 6.6 for 1 month before they were put back to pH_e_ 7.4 for 2 weeks. Experiments requiring pH_e_-selected + 7.4 cells were performed after a maximum of 2 weeks following recovery to pH_e_ 7.4. Both pH_e_ 7.4 and pH_e_ 6.6 media were refreshed every other day, and acidic medium pH was monitored daily. For experiments requiring acute (1 h) exposition to acidic pH, an aliquot of pH_e_ 6.6 medium was maintained at 37 °C under a humidified 5% CO_2_ atmosphere for at least 1 h before the experiment to equilibrate the target pH_e_.

### 2.2. Cell Proliferation and Viability

Proliferation of PANC-1 and Mia PaCa-2 cells was studied using EdU staining assay, MTS assay, trypan blue exclusion assay, and ATP quantification assay. For MTS assays, cells were plated in 96-well plates at a density of 3000 cells/well for PANC-1 and 8000 cells/well for Mia PaCa-2 cells in 100 µL medium, letting them adhere overnight. Cell media of correspondent pH was refreshed the following day and cell proliferation was assessed at 2 h, 24 h, 48 h, 72 h, and 96 h using MTS CellTiter 96^®^ AQueous assay (Promega (Madison, WI, USA); Cat# G1111) following the manufacturer’s instructions. 

Cell proliferation was assessed via 5-ethynyl-2′-deoxyuridine (EdU) incorporation using Click-iT EdU Alexa Fluor 647 Imaging Kit (Invitrogen; Cat# C10340). Briefly, the different PANC-1 and Mia PaCa-2 cell models were plated for 4 days in their corresponding pH_e_ conditions in 1% gelatin-coated coverslips inserted in 6-well plates. The culture medium was refreshed every two days, and cells were incubated with 20 µM EdU for 2 h at 370 °C. Cells were then washed with phosphate-buffered saline (PBS; pH 7.4; Gibco; Cat# 10010-015) and fixed with 4% paraformaldehyde (PFA) at room temperature for 15 min. Following two washes with 3% bovine serum albumin (BSA)-containing PBS, cells were permeabilized with 0.5% Triton X-100 in PBS at room temperature for 20 min and washed twice with 3% BSA-containing PBS. Cells were then stained with 100 µL Click-iT reaction cocktail per coverslip and incubated in the dark for 30 min. Following two washes with 3% BSA-containing PBS, cell nuclei were stained with Hoechst 33342 for 30 min and away from light. Coverslips were finally mounted on clean, microscope glass slides using Glycergel mounting medium (Dako (Santa Clara, CA, USA); Cat# C0563) and stored in the dark at 4 °C. The EdU-positive cells were visualized using a confocal laser scanning microscope (LSM 700; Carl Zeiss MicroImaging GmbH, Oberkochen, Germany) with a Plan Apochromat 40×/1.3 numerical aperture oil-immersion objective. At least five photos per condition were taken, and positive cells were counted using ImageJ software (v1.54d). 

Trypan blue exclusion assays were used to determine the viability of PANC-1 and Mia PaCa-2 cell models after exposition to pH_e_ 6.6 for different time points (2 h, 24 h, 48 h, 72 h, and 96 h, 15 days, and 30 days). PANC-1 and Mia PaCa-2 cells were seeded in 6-well plates at a density of 2 × 10^5^ cells/well and allowed to adhere overnight. Cells were then treated with the acidic medium for the different time points. Cells were trypsinized at specific time points and stained with trypan blue to assess the live/dead cell counts following counting on a hemocytometer. Data from treated cells with acidic pH_e_ were compared to those from PANC-1 and Mia PaCa cells kept in pH_e_ 7.4. Three independent experiments were performed for each experimental condition.

### 2.3. Cell Adhesion

Adhesion assays quantified the ability of PDAC cells to remain adhered to 1% gelatin-coated wells when exposed to a detachment force. PANC-1 and Mia PaCa-2 cells were plated in 96-well plates at a density of 3000 cells/well, and 6000 cells/well, respectively, in 100 µL of the appropriate growth medium containing 10% FBS. PANC-1 and Mia PaCa-2 were incubated at 37 °C and 5% CO_2_ for 1 and 2 h, respectively, to allow the cells to adhere to the well surface. Each condition of each independent experiment was performed in eight technical replicates. Media was discarded, and non-adherent cells were further removed with two washes with cold PBS containing Ca^2+^ and Mg^2+^. Adherent cells were fixed with cold methanol for 15 min at 4 °C, followed by two washes with cold PBS containing Ca^2+^ and Mg^2+^. Nuclei were stained with DAPI in PBS for 15 min at room temperature, and cells were then washed and kept in PBS. Image acquisition was performed using a Nikon Eclipse Ti fluorescence microscope (Nikon Corporation, Tokyo, Japan) with a 4× objective. Cell nuclei of adherent cells of 4 different fields per well were counted in ImageJ for a total of 32 images per condition/biological replicate. The mean numbers of adherent cells obtained from the 32 images of each biological replicate were used for statistical analysis. Three independent experiments were performed for each condition tested. 

### 2.4. Cell Migration and Cell Invasion

Transwell migration and invasion assays were performed using 6.4 mm cell culture inserts with 8 µm pore-size polyethylene terephthalate (PET) membrane (Corning (New York, NY, USA), Cat# 353097) placed into Falcon^®^ 24-well Permeable Support Companion Plate (Corning, Cat# 353504). Cells were pretreated with 10 μg/mL of cell cycle inhibitor mitomycin C for 2 h before seeding. The lower chamber was filled with 500 μL of growth medium pH_e_ 7.4 or 6.6 containing 10% FBS. The 75 × 10^3^ PANC-1 and 10 × 10^4^ Mia PaCa-2 cells/insert were seeded without Matrigel coating (migration assay) or with Matrigel (diluted in growth medium in ratio 1:5; Corning; Cat# 354230) coating in 300 µL of the corresponding pH_e_-target growth medium supplemented with 10% FBS. Thereby, we avoided a FBS gradient between the two compartments. The cells were allowed to migrate or invade through the pores of the insert membrane overnight (around 18 h) at 37 °C and 5% CO_2_. 

Regarding pH gradient experiments, one major limitation of the Transwell system is the inability to maintain a stable pH gradient throughout the experiment (18 h). The pH gradient established by this system is transient due to the diffusion of the acidic medium to the lower compartment with pH_e_ 7.4, which dissipates the pH gradient. Measurements of the medium pH present in the upper chamber revealed that the pH gradient along the Transwell system starts to decrease within ~5 h of the experiment in the absence of the Matrigel coating, while a substantial dissipation of the pH gradient occurred after ~7 h in the presence of the Matrigel coating. For these reasons, experiments with a pH gradient were performed in a 5-h time window, allowing the pH gradient to remain present (Appendix A). After incubation, cells were washed with PBS, and non-migrating/invading cells were removed from the upper side of the membrane using a cotton swab; cell fixation with cold methanol for 15 min at −20 °C was then performed. Cells were stained with 0.4% crystal violet solution at room temperature in the dark for 30 min and imaged under a light microscope (10× magnification). Invasive cells were counted in five representative fields of view per condition. The mean numbers of cells obtained from the five images of each biological replicate were used for statistical analysis. Three independent experiments were performed for each experimental condition.

### 2.5. Immunofluorescence Staining 

PANC-1 and Mia PaCa-2 cells were stained with fluorescent phallotoxin to label F-actin for cell morphology studies. Cells were plated in their corresponding pH_e_ conditions in 1% gelatin-coated coverslips inserted in 6-well plates at different densities to reach ~80% confluence on the day of the experiment. Cells were washed twice with PBS and fixed with 4% paraformaldehyde for 10 min at room temperature. After washing the samples twice with PBS, the fixed cells were incubated with a solution of 0.1% Triton X-100 in PBS (PBST) for 5 min at room temperature for permeabilization. Solutions were decanted, and two washes with PBS followed. Fixed cells were incubated with PBS containing 1% BSA for 30 min and then stained with a staining solution containing 2 units/coverslip of fluorescent Alexa 488 phalloidin (2 U/200 µL per coverslip; Invitrogen™; Cat# A12379) in 1% BSA in PBS for 20 min at room temperature. Cells were washed two times with PBS, and cell nuclei were stained with a solution of 1:1000 DAPI in PBS for 15 min at room temperature. Cells were washed twice with PBS and air dried, and the coverslips were mounted on clean microscope glass slides using Glycergel mounting medium (Dako; Cat# C0563) and stored in the dark at 4 °C. Glass slides were then examined using a confocal laser scanning microscope (LSM 700; Carl Zeiss MicroImaging GmbH) with a Plan Apochromat 40×/1.3 numerical aperture oil-immersion objective. The images were analyzed in Zeiss LSM Image Browser software.

### 2.6. Analysis of Cell Morphology: Cell Area and Cell Circularity

Cells were plated in the appropriate pH_e_-target medium and grown to approximately 80% confluency. Four random fields of each cell condition were imaged using a brightfield microscope (Nikon Eclipse TS100). ImageJ software quantified the cell area and circularity by manually segmenting the brightfield images. Over 500 cells for each condition were considered for the cell area and circularity analysis.

### 2.7. RNA Extraction and qPCR

Total RNA was isolated from cultured PANC-1 and Mia PaCa-2 cell models using the NucleoSpin RNA Plus kit (Macherey Nagel Bioanalysis™, Bethlehem, PA, USA), according to the manufacturer’s instructions. RNA concentration and quality were determined by absorbance at 260 nm. Reverse Transcription synthesized cDNA in a 20-μL reaction mixture containing 2 μg total RNA, 10 mM dNTPs, 50 µM/100 µL Random Hexamers, 1× First strand buffer, 20U RNase inhibitor, 0.1 M DTT, and 200 U M-MLV reverse transcriptase. Real-time qPCR was performed in a 20 µL-reaction mixture containing 10 ng/µL cDNA, SYBR™ Green PCR Master Mix (ThermoFisher Scientific), and 400 nM forward and reverse primers for each gene of interest. qPCR reactions were performed in technical triplicates. The primers used are listed in Table 1. The analysis was performed in a real-time thermal cycler Cfx C1000 (Bio-Rad, Hercules, CA, USA). Hypoxanthine Phosphoribosyltransferase-1 (HPRT-1) was used for normalization. Relative mRNA levels were quantified using the 2(−ΔΔCT) method.

### 2.8. Protein Extraction and Western Blot

PANC-1 and Mia PaCa-2 cells were washed with ice-cold PBS and lysed in RIPA buffer (1% Triton X-100, 0.1% sodium deoxycholate, 150 mM NaCl, 10 mM PO_4_Na_2_/K, pH 7.4) containing protease and phosphatase inhibitor cocktails (Sigma-Aldrich and Thermo Scientific™, respectively). Protein concentrations were determined with the Bicinchoninic Acid protein assay (Thermo Fisher Scientific, Waltham, MA, USA), and 30 µg of denatured protein lysate was used for each condition. Samples were loaded in sodium dodecyl sulfate-polyacrylamide gels (7 or 12%) of 1.5-mm thickness and electrophoresed in tris-glycine migration buffer (25 mM tris base, 192 mM glycine, 0.1% SDS, pH 8.3–8.5) at 80 V in stacking gel and 120 V in resolving gel. Protein samples were transferred from the polyacrylamide gel onto a nitrocellulose membrane using a Pierce G2 Fast Blotter System (Thermo Scientific) at 2.5 V and a 3 A constant for 15 min. Nitrocellulose membranes were blocked with 3% BSA in 1X TNT buffer (15 mM Tris-HCl, 140 mM NaCl, 0.05% Tween 20, pH 7.5) for 1 h at room temperature followed by an overnight incubation at 4 °C with the following specific primary antibodies preparing in 3% BSA in 1X TNT supplemented with 0.02% sodium azide: anti-E-cadherin (1:1000, Cat# MAB3199), anti-N-cadherin (1:200, Cat# SC59987), anti-Vimentin (1:1000, Cat# SC5565), anti-Actin (1:1000, Cat# A5441), anti-Calnexin (1:1000, Cat# MAB3126). Following overnight incubation, three washes of 5 min each and one wash of 10 min in 1X TNT were performed before membranes were incubated with 3% BSA in TNT solution containing goat anti-rabbit IgG (Jackson ImmunoResearch (West Grove, PA, USA); Cat# 211-032-171, 1:50,000) or goat anti-mouse IgG (Jackson ImmunoResearch; Cat# 115-035-174, 1:25,000) horseradish peroxidase-conjugated secondary antibodies for 1 h at room temperature. Membranes were then washed for 3 × 5 min and 1 × 10 min in 1X TNT. Peroxidase activity was revealed using SuperSignal West Dura or SuperSignal West Femto chemiluminescent substrate (Thermo Fisher Scientific, Waltham, MA, USA), according to the manufacturer’s instructions. Chemiluminescent signals were captured on Amersham Imager 600 (GE Healthcare Life Sciences, Chicago, IL, USA) and quantified using the densitometric analysis option in ImageJ/Fiji version 1.53 software. All band density values were normalized to β-actin or Calnexin, used as loading controls, and then compared to the control condition.

### 2.9. Intracellular pH Measurements

The intracellular pH of PANC-1 cells was measured using fluorescent live-cell imaging (Axiovert TV100, Zeiss, Oberkochen, Germany) as described previously [45]. Cells were loaded with the fluorescent pH indicator 2′7′-bis(carboxyethyl)-5-carboxyfluorescein (BCECF-AM) (3 µM) for up to 2 min. The excitation wavelength alternated between 490 nm and 440 nm. The emitted fluorescence was detected at 510 nm. The mean fluorescence of each cell was measured in 10 s intervals. Data acquisition and the polychromator (Visitron Systems, Puchheim, Germany) were controlled by the program VisiView (Visitron Systems). The cells were superfused with prewarmed (37 °C) CO_2_/HCO_3_^−^-buffered Ringer’s solution (116 mM NaCl; 24 mM NaHCO_3_; 5.4 mM KCl; 0.8 mM MgCl_2_; 1.2 mM CaCl_2_; 5.5 mM Glucose) at pH 7.4. NaHCO_3_ was lowered to 4.7 mM for pH 6.6. pH measurements were calibrated with a two-point calibration (130 mM KCl; 1.2 mM CaCl_2_; 0.8 mM MgCl_2_; 10 mM Hepes; 5.5 mM Glucose; pH 7.5 and pH 6.5; supplemented with 10 µM Nigericin) (Sigma-Aldrich, Merck KGaA, Darmstadt, Germany). For data analysis, the mean fluorescence intensity of the cell area was measured and corrected for background fluorescence. Afterward, the 490 nm/440 nm ratio was determined, and the pH_i_ was calculated with a linear regression of pH 6.5 and 7.5.

### 2.10. Invadopodia Activity Assay: Fluorescent-Matrigel Layer Preparation and ECM Digestion Index Assay

Invadopodia focal ECM digestion experiments were conducted as previously described [46]. Cells were seeded onto a layer of 90% matrigel:10% collagen I (3.6 mg/mL Matrigel and 0.4 mg/mL collagen I) in which quenched BODIPY linked to BSA (DQ-Green-BSA) was mixed at a final concentration of 30 µg/mL. The matrix mix was used to cover 12-mm round glass coverslips at the bottom of a 24-well plate. Matrigel containing the fluorescent dye was allowed to polymerize for 30 min in a humidified incubator at 37 °C. Then, 30 × 10^3^ cells/coverslip were seeded on the top of polymerized ECM at both pH_e_ 7.4 and pH_e_ 6.6 and incubated overnight. Cells were fixed with paraformaldehyde 3.7% in PBS, stained for F-actin with Phalloidin–Tetramethylrhodamine B isothiocyanate (1:5000 in 0.1% gelatin in PBS, Sigma-Aldrich Cat# p1951), and processed for immunofluorescence. Invadopodia-dependent ECM digestion was evaluated microscopically. Invadopodia-ECM digestion emits green fluorescence on a black background, which quantitatively reflects their ECM proteolytic activity. The quantity of invadopodia ECM proteolysis for 100 cells was then calculated as follows:

Digestion Index = % of cells positive for invadopodial ECM digestion × mean pixel density of focal ECM digestion/cell.

### 2.11. RNA-Sequencing and Analysis

Total RNA was isolated from cultured PANC-1 cells using the NucleoSpin RNA Plus kit (Macherey Nagel Bioanalysis™), according to the manufacturer’s instructions. Libraries for the RNA-Seq analysis were prepared with 200 ng of RNA using the Illumina Stranded mRNA Prep Kit (Illumina, Inc., San Diego, CA, USA), according to the manufacturer’s protocol. All the libraries were sequenced on an Illumina NovaSeq 6000 system using paired-end NovaSeq 6000 S2 Reagent Kit (200 cycles). Raw sequencing reads were first trimmed using BBDuk (from BBTools suite v35.85) to remove possible adapter contamination and then aligned to the reference genome (Ensembl Release 106) using STAR aligner v2.5 [47]. RSEM v1.3.0 [48] was used to quantify transcript abundances, while differential expression analysis was carried out using the package DESeq2 v1.14.1 [49] in R/Bioconductor environment. False discovery rate (FDR) [50] was controlled at level α = 0.05 through the Benjamini–Hochberg procedure. Genes with an adjusted *p*-value smaller than 0.05 and featuring a |log_2_FC| > 1 were considered significantly deregulated. The ToppFun web tool (from Topmen Suite, https://toppgene.cchmc.org/, accessed on 4 August 2022) was used to identify significantly overrepresented GO terms and pathways [51]. Finally, log_2_ fold changes determined with DESeq2 were used to rank genes for Gene Set Enrichment Analysis (GSEA v4.2.2) with the MSigDB database v2022.1.Hs (updated August 2022) [52]. Probes were collapsed into unique gene symbols before the analysis, and a standard (weighted) enrichment statistic was chosen for the computation of the Normalized Enriched Score (NES). Within the context of the GSEA, the *q*-values cutoff was set to 0.25.

### 2.12. Statistical Analyses

The data were analyzed using GraphPad Prism 7 software (GraphPad Corporation, San Diego, CA, USA). The Shapiro–Wilk normality test was used to assess the normality of distribution of the continuous variables, which were reported as mean and standard deviation (SD) or standard error of the mean (SEM). In contrast, non-normally distributed variables were reported as a median and 95% Confidence Interval. The Student’s *t*-test was used to compare the means of two continuous variables with normal distribution. In contrast, the Mann–Whitney *U* test was used for non-normal distributed variables. Means of more than two groups of variables were compared using One-way ANOVA or the Kruskal–Wallis *H*-test. A *p*-value < 0.05 was considered significant.

## 3. Results

### 3.1. Extracellular Acidification Selection Affects PANC-1 Intracellular pH and PDAC Cell Line Morphology

The role of an acidic pH_e_ in determining phenotypic changes was assessed in PANC-1 and Mia PaCa-2 pancreatic cancer cell lines. They are characterized by a mesenchymal phenotype and are poorly differentiated [53]. We chose pH_e_ 6.6 which is in the typical pH_e_ range of solid tumor areas (pH_e_ 6.4 to 7.2), including pancreatic cancer [54]. The cells were subjected to pH_e_ 6.6 for a short-term period («4 days pH_e_ 6.6» cells) or long-term period («pH_e_-selected» cells) to simulate and study the early and late stages of low pH_e_ selection (see also the Materials and Methods section for details) (Figure 1a). Moreover, to mimic the heterogeneity of the tumor pH landscape and, in particular, the tumor edges at the interface with peritumoral tissue, PDAC cell lines were recovered at pH_e_ 7.4 after pH selection («pH_e_-selected + 7.4» cells) (Figure 1a). 

The first step in evaluating the role of the extracellular acidic tumor microenvironment was to monitor its effect on intracellular pH (pH_i_). It is a key parameter for all types of cells, profoundly affecting several cell processes, including those promoting cancer progression [7]. This value does not always correspond to the extracellular pH_e_ value in cancer cells but is influenced by it. Figure 1b shows the mean traces of PANC-1 cells’ resting intracellular pH measurements, obtained by superfusing each cell model with CO_2_/HCO_3_^−^-buffered Ringer’s solution at the appropriate pH_e_ for 3 min. Extracellular acidification at pH_e_ 6.6 treatment slightly influences the intracellular pH that became more acid compared to the control condition, although the difference is not statistically significant (pH_i_ 6.99 ± 0.12 control conditions; pH_i_ 6.72 ± 0.10 4 days pH_e_ 6.6; pH_i_ 6.57 ± 0.21 pH_e_-selected cells). In contrast, recovery to pH_e_ 7.4 following the 1-month-long pH_e_ 6.6 treatment determines a significant cytosolic alkalinization compared to pH_e_-selected cells, although not statistically significant compared to control cells (*p*-value = 0.50) (pH_i_ 7.31 ± 0.17) (Figure 1b,c). These results indicate that extracellular acidosis leads to a mild intracellular acidification, affecting PANC-1 cells exposed to acidic pH_e_ for extended periods. As opposed to this phenomenon, 1-month acidic treatment followed by a pH_e_ 7.4 exposition recovers and alkalinizes the pH_i_.

To examine the role of short- or long-term acidification on PDAC cell lines, we evaluated the morphology of PANC-1 and Mia PaCa-2 cells after labeling the cells with Alexa 488 phalloidin and DAPI (Figure 1d–f). Both cell lines display a more spread-out morphology in the initial days of the acidic pH_e_ exposure (4 days pH_e_ 6.6), compared to control cells (Figure 1d). This is accompanied by an increase in cell area and a decrease in cell circularity (Figure 1e,f, green bars). This spread-out and elongated shape is still maintained in PANC-1 and Mia PaCa-2 selected cells + 7.4 (Figure 1d–f, red bars). Interestingly, Mia PaCa-2 cells show similar behavior after 1-month of acid selection (Figure 1d,f, pink bars). This treatment induces a significant decrease in the cell area of PANC-1 cells (Figure 1d,e, pink bar).

### 3.2. Extracellular Acidification Decreases Cell Proliferation in PANC-1 and Mia PaCa-2 Cells

Cell proliferation is strictly regulated by the pH_e_/pH_i_ ratio. We quantified DNA-synthesizing cells by using EdU incorporation, and the metabolic activity as an indication of the proliferative state of cells was quantified by MTS assay. The data showed a significant inhibition of both PANC-1 and Mia PaCa-2 cell proliferation by extracellular acidification independently of the time of acidic pH_e_ exposure. Indeed, the ratio of Edu positive/Hoechst cells is reduced and the absorbance of the MTS reagent is reduced in 4 days pH_e_ 6.6 and pH_e_-selected cells when compared to the control groups (Figure 2a–f, Appendix A). Data were also confirmed by quantifying the ATP production (Appendix A). Interestingly, a significant increase in the fraction of EdU-positive cells or increase in MTS absorbance is detected in pH_e_-selected + 7.4 cells, demonstrating that pH_e_ 7.4 treatment following long-term acid exposure enhances cell proliferation in both PDAC cell lines as compared to control conditions (Figure 2c–f).

The enhancement in cell viability in pH_e_-selected cells after recovery at pH_e_ 7.4 is further confirmed in PANC-1 by quantification of the ATP production by metabolically active cells. However, no significant differences are detected between Mia PaCa-2 control and pH_e_-selected + 7.4 (Appendix A). The reduced proliferation observed in PANC-1 and Mia PaCa-2 cells following a 4 days-long treatment with acidic pH_e_ is the consequence of an increase in cell death, as detected by the trypan blue exclusion assay. The latter shows that the percentage of dead cells reaches almost 50% after 4 days in acidic pH_e_ media for PANC-1 cells and nearly 30% in Mia PaCa-2 cells (Figure 2g). However, this phenomenon appears to be overcome as the acidic treatment continued. After 15 days-long acidic treatment, the percentage of live cells recovers, reaching almost 90% after 1 month in low pH_e_ conditions (Figure 2h). These results indicate that acute acidification promotes significant cell death in both PDAC cell lines (Figure 2h) within the first 96 h after seeding, selecting a subpopulation of cells that outgrows and is viable under acidic conditions, with a higher proliferation rate as compared to non-selected PANC-1 cells (Figure 2e,* and $ symbol), which is consistent with more aggressive phenotype. To better clarify the acquired proliferative and more acid-resistant behavior of PDAC pH_e_-selected + 7.4 cells, they were subjected to a 4-days long treatment with pH_e_ 6.6. Cell viability was assessed at the end of the treatment, obtaining no significant differences in the percentage of live cells when compared with the same cells kept in pH_e_ 7.4 and control cells (Figure 2i). These results indicate that PDAC control and pH_e_-selected + 7.4 cells respond differently to low pH_e,_ thus suggesting a selection of acid-resistant cells whose phenotype is also maintained after recovering in pH_e_ 7.4.

### 3.3. Acid Selection at pH_e_ 6.6 Promotes Adhesion, Migration, and Invasion of PANC-1 and Mia PaCa-2 Cells 

The role of acidic pH_e_ on cell migration and invasion was initially investigated by studying their ability to adhere to gelatin. A period of 4 days of extracellular acidification significantly inhibits both PANC-1 and Mia PaCa-2 cell adhesion compared to the control condition (~50 % reduction; Figure 3a,b). The inhibitory effect of acidic pH_e_ is already present after 1 h of acidic exposure of PANC-1 cells growing at pH_e_ 7.4 (control cells) (Appendix A). On the contrary, one-month extracellular acidification (pH_e_-selected cells) promotes an increase in PANC-1 and Mia PaCa-2 adhesion compared to their respective control groups. A similar positive effect on cell adhesion is observed on adhesion of PANC-1 pH_e_-selected + 7.4 cells (Figure 3a). However, surprisingly Mia PaCa-2 pH_e_-selected + 7.4 cells decrease their adhesion ability as compared to control (Figure 3b). Mia PaCa-2 cells are less sensitive to a 1 h acid treatment, which does not affect the adhesive properties of pH_e_-selected cells (Appendix A).

In the second set of experiments, the migratory ability of PDAC cells was evaluated using Transwell migration assays. The data presented in Figure 3c–e show that cell migration is significantly enhanced during the early stages of pH_e_ treatment (4 days pH_e_ 6.6) and in pH_e_-selected cells, with a maximal effect in pH_e_-selected + 7.4 (Figure 3c–e). Time-lapse video-microscopy assays were also performed to confirm the enhanced migratory activity at the single-cell level after 4 days of acidic pH_e_ treatment in PANC-1 cells (Appendix A). This increase in PANC-1 cell migration nicely correlates with the higher recruitment of paxillin, a central Focal Adhesion (FA)-adaptor protein and a marker for nascent focal adhesion [55] toward the cell periphery and by larger FAs compared to control (Appendix A). 

The correlation between migratory cell properties and cell invasion was studied by using Matrigel-coated Transwell filters. Interestingly, early stages of acidic selection (4 days pH_e_ 6.6) significantly inhibit the invasive abilities of both PANC-1 and Mia PaCa-2 cells (Figure 3f–h). The effect is reversed with a prolonged exposition to acidic pH_e_ conditions (pH_e_-selected cells) at a similar rate as pH_e_-selected + 7.4 cells (Figure 3f–h). 

The role of pH_e_ selection on both migration and invasion in the context of a microenvironmental pH_e_ gradient was evaluated by experimentally establishing a pH_e_ gradient in the Transwell system. This experimental condition would mimic the interface between acidic cancer edges and neighboring healthy tissues and blood vessels. To simulate this situation, PDAC cells were plated in pH_e_ 6.6 media and allowed to migrate or to invade (through the Matrigel coating) toward pH_e_ 7.4 media overnight. Migration assay clearly demonstrates that all three acid-treated PDAC cell models (4 days pH_e_ 6.6; pH_e_-selected and pH_e_-selected + 7.4) respond to the pH_e_ gradient more avidly than the control cells (Figure 4a–c). In contrast, invasion of PANC-1 and Mia PaCa-2 cells exposed to pH_e_ 6.6 for a short term (4 days pH_e_ 6.6 cells) along the alkalinizing pH_e_ gradient is significantly inhibited compared to control cells (Figure 4d–f). On the contrary, prolonged treatment with acidic media enhances the invasion of both PDAC pH_e_-selected cells towards the compartment filled with pH_e_ 7.4 as compared to control. This invasive behavior is maintained in PANC-1 and Mia PaCa-2 pH_e_-selected + 7.4 cells (Figure 4d–f).

Cancer cells digest extracellular matrix (ECM) on their way to local or distant metastases. To accomplish this task, cells employ specialized structures called invadopodia, F-actin-rich membrane protrusions that mediate protease-dependent proteolysis of ECM components. Previous works have elucidated the critical role of extracellular acidosis in invadopodial function and invasion [46,56,57,58,59,60]. Therefore, we next assessed whether the extracellular acidosis could promote the invadopodia-mediated proteolytic degradation of the ECM by performing an in situ zymography in both PANC-1 and Mia PaCa-2 cells. To this purpose, cells were plated on quenched DQ-Green-BSA-containing Matrigel-coated slides the invadopodia-mediated focal ECM digestion was quantified via the measurement of the ECM Digestion Index (Figure 4g,h), which derives from the combination of % of cells positive for invadopodial ECM digestion x mean pixel density of focal ECM digestion/cell (see Section 2.10) that are presented as single parameters in Appendix A. The invadopodia-mediated ECM digestion assay was performed in both PDAC control and pH_e_-selected + 7.4 cells subjected to an overnight exposition to acidic pH_e_ (pH_e_ 6.6) or physiological pH_e_ (pH_e_ 7.4) (same protocol as in Figure 4a–f). Acidic overnight treatment (pH_e_ 6.6) of both PANC-1 and Mia PaCa-2 control cells reduces the invadopodial Digestion Index compared to control cells exposed to pH_e_ 7.4 (Figure 4g,h). On the contrary, the Digestion Index is very much higher in both PANC-1 and Mia PaCa-2 pH_e_-selected + 7.4 cells compared to the control cells at pH_e_ 7.4 and pH_e_ 6.6. Moreover, extracellular acidosis further increases the Digestion Index of pH_e_-selected + 7.4 cells compared to the same cells at physiological pH_e_. Thus, prolonged exposure of PDAC cells to acidic conditions selects cells with higher proteolytic invadopodial activity, conferring a more aggressive and invasive phenotype to these cells. 

### 3.4. Effects of pH_e_ 6.6 on Epithelial-Mesenchymal Transition and Proliferation Markers of PANC-1 and Mia PaCa-2 Cells

Considering the previous results and the more aggressive behavior showed by acid-selected cells, we assessed whether acidic pH_e_-induced selection determines the acquisition of mesenchymal properties in PDAC cell lines. Therefore, different EMT markers were evaluated at mRNA levels by qPCR. As previously demonstrated [53], PANC-1 and Mia PaCa-2 cell lines are characterized by low E-cadherin mRNA expression levels, which are not affected by the different acidic pH_e_ exposures (Figure 5a,b and Appendix A).

E-cadherin and N-cadherin mRNA expression levels were then validated at the protein level in both cell lines (Figure 5d,e). Although the PANC-1 cell line shows increased N-cadherin expression during the different exposures to acidic pH_e_ at mRNA level (Figure 5a,b), it reaches significant protein upregulation only in pH_e_-selected + 7.4 cells (Figure 5d,e). Mia PaCa-2 cells follow a similar mRNA expression pattern in the different acidic conditions (Figure 5b). The acidic treatment-induced shift towards the acquisition of a more mesenchymal phenotype in PANC-1 and Mia PaCa-2 cells is confirmed by the increased mRNA expression of additional mesenchymal markers, Twist1 and Snail. (Figure 5a,b), while Vimentin mRNA levels are significantly increased in PANC-1 cells (Figure 5a,b) but not in Mia PaCa-2 cells, as confirmed by protein quantification (Figure 5d,g). Vimentin overexpression is confirmed by Western blot, showing a significant increase in PANC-1 cells exposed to 1 month to pH_e_ 6.6 and those subsequently recovered to pH_e_ 7.4 (Figure 5d,f). Both PDAC cell lines express Slug at low mRNA levels (Appendix A). However, its gene downregulation is detected following 4 days-long acidic treatment (Figure 5a,b), while a clear upregulation emerges in pH_e_-selected cells recovered to pH_e_ 7.4. These results reinforce the idea that the acidic tumor microenvironment promotes the acquisition of more aggressive cellular phenotypes via the epithelial–mesenchymal transition, especially in PANC-1 cells.

To further support the previous proliferation results (Figure 2 and Appendix A), PANC-1 and Mia PaCa-2 cells were tested for the mRNA expression of Ki67, an important proliferation marker, and G0S2, a protein involved in the maintenance of hematopoietic stem cells in a quiescent state [61] (Figure 5c). Ki67 upregulation is confirmed in PANC-1 and Mia PaCa-2 pH_e_-selected + 7.4 cells, indicating their enhanced proliferative potential compared to control cells, while its expression is decreased in 4 days- and 1-month-long acidic treatment (Figure 5c). Both PDAC cell lines exposed for 4 days to pH_e_ 6.6 then display a proliferation arrest in the G0 phase, as demonstrated by the enhanced mRNA expression of the G0S2 marker. The reversibility of this cell-cycle arrest is shown by the progressive decrease in the G0S2 marker expression in cells following 1-month-long pH_e_ 6.6 treatment and further recovered to physiological pH_e_, particularly in PANC-1 cells (Figure 5c).

### 3.5. Differential Transcriptomic Profiles in PANC-1 Cells in Response to Acidosis

To study the effects of the acidic pH_e_ on gene expression profile in PANC-1 cells, RNA samples from control, 4 days pH_e_ 6.6, and pH_e_-selected + 7.4 cell models were subjected to RNA-sequencing.

Raw paired-end reads were first trimmed to remove adapter contamination; then, STAR aligner was used with RSEM to obtain the count table at gene level as detailed in the Materials and Methods section. Principal component analysis (PCA) was conducted to check the consistency of the three experimental groups (Appendix A); then, the DESeq2 Bioconductor package was used to assess differential gene expression. Our results lead to the identification of 772 differentially expressed genes (DEGs) in PANC-1 cells exposed for 4 days to pH_e_ 6.6 compared to control samples (398 genes downregulated and 374 genes upregulated), while 989 DEGs are identified between control and pH_e_-selected + 7.4 samples (530 genes downregulated and 459 genes upregulated) (Figure 6a–d and Appendix A for the full DEG lists and the related statistics). Notably, just 134 genes (8.2% of the total DEGs) are in common between the two experimental conditions of pH_e_ manipulation (see Venn diagram in Figure 6e), representing the transcriptional correlate of that specificity already observed through functional assays. 

Highly upregulated genes in 4 days pH_e_ 6.6 cells includes GBP3, a member of the guanylate-binding protein family, which induces caspase-dependent apoptosis in leukemia cells [62] and exerts an anti-tumor role in colorectal cancer [63]. However, its pro-tumor role has also been previously reported [64]. Among the top five downregulated genes in 4 days pH_e_ 6.6 cells, we find PXDN, coding for Peroxidasin, a heme peroxidase which has been associated with proliferation in endothelial cells [65] and with cancer cells invasive phenotype [66,67], Guanylate Cyclase 1 Soluble Subunit Alpha 2 coding gene (GUCY1A2), which mediates cell growth and survival in different cancer cell types [68,69,70,71,72], and PRICKLE1, prickle planar cell polarity protein 1, a member of the planar cell polarity (PCP) pathway which is involved in cancer cell metastasis [73,74]. Stronger gene deregulations are observed for pH_e_-selected + 7.4 cells, where the top 10 upregulated genes include Interferon Alpha Inducible Protein 27 (IFI27), a prognostic marker for pancreatic cancer [75] that promotes PDAC cell proliferation, migration, and invasion [76]; bone morphogenetic protein 7 (BMP7), a member of the transforming growth factor-β (TGF-β) family reported to be involved in tumor metastasis, including pancreatic cancer where it promotes EMT and invasiveness in PANC-1 cells via matrix metalloproteinase (MMP)-2 upregulation [77]; Interleukin-32 (IL32), involved in PDAC cells invasiveness [78]; tripartite motif-containing 2 (TRIM2), which increases PDAC tumorigenesis both in vitro and in vivo [79]. Among top downregulated genes, amphiregulin (AREG) is involved in EMT and growth in different types of cancer, including PDAC [80], while Adhesion G Protein-Coupled Receptor F1 (ADGRF1) has an important role in inducing quiescence and chemoresistance in breast cancer [81]. Other downregulated genes include Traf2- and Nck-interacting kinase (TNIK), involved in colorectal carcinogenesis via modulation of Wnt signaling pathway [82], and Ankyrin-3 (ANK3), whose knock-down was reported to decrease the growth of prostate cancer cells while promoting their invasion both in vitro and in vivo [83], and to inhibit invasive abilities of thyroid cancer cells and their tumorigenesis when ectopically expressed [84].

To deepen the role of pH_e_ selection and recovery in EMT process, we performed a hypergeometric test to measure the significance of the overlap between our DEG list and the set of genes involved in EMT as provided by dbEMT 2.0 (http://dbemt.bioinfo-minzhao.org) [85]. Interestingly, the data clearly show a 1.7-fold enrichment of the EMT gene set in our DEG list (*p*-value = 7.27 × 10^−7^) in pH_e_-selected + 7.4 cells as well as in 4 days pH_e_ 6.6 cells (*p*-value = 9.91 × 10^−6^, fold enrichment = 1.7) (Figure 7a,b; see Appendix A for a complete list of EMT DEGs). These data agree with the significant increase in EMT markers expression in both early and late stages of acidosis adaptation (Figure 5), thus supporting the hypothesis that acidosis selects more invasive cell phenotypes by promoting EMT and thus paving the way for more aggressive cell phenotypes.

To corroborate the functional assays and results obtained in vitro, up- and downregulated DEG lists were separately tested for over-representation through hypergeometric distribution by ToppFun web tool (see Materials and Methods section). The analysis of 4 days pH_e_ 6.6 downregulated genes returned many enriched GO terms (biological processes) related to cell adhesion, cell growth, proliferation, and (negative regulation of) migration (Figure 7c), whereas very few terms resulted in upregulated PANC-1 cells exposed to short-term acidosis (Figure 7e). Pathway analysis further confirmed the involvement of 4 days pH_e_ 6.6 downregulated genes in cell adhesion-related processes (such as β1 integrin cell surface interactions and focal adhesions) as well as a significant contribution to cholesterol biosynthesis and NOD-like receptor signaling pathway coming from the upregulated genes (Appendix A, respectively). As for the pH_e_-selected + 7.4 condition, GO terms (biological processes) linked to the negative regulation of cell cycle and cell growth are significantly over-represented within the set of the downregulated genes (Figure 7d), while biological processes associated with cell adhesion, cell migration, and EMT are significantly enriched in the upregulated gene list (Figure 7f). Accordingly, pathway analysis points at a significant alteration in cell cycle-associated processes due to the genes downregulated in pH_e_-selected + 7.4 condition, while the upregulated ones are mostly involved in the cadherin signaling pathway, interferon signaling, and cell invasion-associated processes, including degradation of the extracellular matrix and DUB metalloproteases (Appendix A). The complete lists of all the enriched terms returned by such an over-representation analysis are provided in the Appendix A.

Finally, to validate and deepen these results, the GO database was also interrogated using GSEA software (see Section 2.11 for more details on parameter settings). GSEA results are in good agreement with the previous ones, showing that 4 days exposure of PANC-1 cells to pH_e_ 6.6 decreases cell adhesion properties (Figure 7g). The long-lasting selection protocol induces a general enhancement of cell adhesion, proliferation, and migration in relation to the mesenchymal-like phenotype of PANC-1 cells kept under acidic conditions (Figure 7h). The full tables of significant terms resulting from GSEA can be found in Appendix A.

## 4. Discussion

In the same way as most other cancers, Pancreatic Ductal Adenocarcinoma is characterized by a highly acidic microenvironment resulting from the aberrant blood perfusion and the metabolic rearrangement that cancer cells put in place to meet the increased demand for energy and nutrients needed for their growth. In this study, we characterized the response of PDAC cell lines, PANC-1 and Mia PaCa-2, to short term (4 days) or long-lasting (1 month) acidosis (pH_e_ 6.6) exposure on PDAC cellular hallmarks. Although previous research already showed that prolonged treatment induce a phenomenon of acid adaptation in pancreatic cells [41,43,86,87] little is known about the metastatic potential of the acid-selected cancer cells. To this purpose we established a model by re-exposing acidic selected cells to pH_e_ 7.4 for two weeks. This treatment mimics the interface between acidic microenvironment of the tumor and the physiological pH_e_ of the adjacent non-tumorous tissue. Indeed, tumor acidosis is not restricted to its hypoxic core, but it extends toward the interface between the tumor and normal tissue which delineates the margin of a malignant tumor when cells invade neighboring healthy tissues and blood vessels [21,88].

Our data indicate that acidic pH_e_ selects for more aggressive pancreatic cancer cells via the increase in cell death during the short-term exposition (4 days). This leads to the survival of acid-resistant cells to low pH_e_ stress that further outgrew and underwent EMT to acquire a more aggressive and invasive phenotype when exposed to pH_e_ 7.4. 

One feature of cancer cells is the pH gradient reversal compared to their healthy counterparts, leading to extracellular acidification coupled with the relative alkalinization of the intracellular space [7,89]. pH_i_ homeostasis is important for the correct functioning of various physiological and pathological cellular processes. Despite the interest of the cancer cell to maintain a stable pH_i_, it is subjected to variations in external pH; therefore, sudden acidification of the extracellular space can result in the disruption of the pH_i_ steady-state, which may pave the way toward selective pressures as a survival mechanism [6,90]. Our results indicated that short and long acidic exposure induces a slight decrease, although not significant, in the PANC-1 basal pH_i_ (Figure 1b,c), in accordance with previous reports [43,91,92]. Our data demonstrated that the basal pH_i_ value is not only recovered but further increased when PANC-1 pH_e_-selected cells face back pH_e_ 7.4 (Figure 1b,c), supporting recent evidences in same cell type [43], a feature that may represent a driving force for cancer progression, as suggested for other cancer types [7,93,94,95]. 

The results on pH_i_ levels correlated well with functional assays, as those PANC-1 cells exposed to acidic pH_e_ for 4 days had both a decrease in the proliferation rate and lower pH_i_ values than control cells. After 1 month of acidic pH_e_ exposure, the growth rate of PDAC cells was comparable to that of cells exposed for 4 days to pH_e_ 6.6. We confirmed this output in Mia PaCa-2 cells, although the inhibition of proliferation was less evident in these cells. Validation of the Ki67 proliferation marker and G0S2 quiescence marker mRNA levels confirmed our proliferation results (Figure 5c). These results agree with the well-established notion that internal acidification, probably due to external acidic pH_e_, inhibits cell proliferation in pancreatic cancer [43,86]. Finally, PDAC cell proliferation was restored and further boosted by recovery in pH_e_ 7.4. These results clearly demonstrate that the acidic tumor microenvironment selects for cells with reduced growth capacities in acidic conditions but which display greater proliferative abilities when they come into contact with areas of physiological pH_e_ [10].

We showed that cell growth inhibition results from an increase in cell death, which gradually increased until it reached the maximum percentage of cell death at 4 days (Figure 2g,h). The reduced cell viability is the result of energy deprivation caused by glycolysis inhibition [96], as demonstrated by ATP shortage (Appendix A) and by intracellular acidification (Figure 1b,c). We observed a subsequent, progressive, and almost complete recovery of cell viability (Figure 2g,h). This is likely due to the selection of a subset of cells resistant to the tumor acidosis-induced toxicity around the middle of the 1-month treatment. Consequently, almost all cells were alive at the end of the 1-month acidic exposure. Such a behavior was also observed in other cancer cell types [24,30,39,43,97].

Potentiation in cell migration and invasion represents a key feature of aggressive cancer progression. Quantification of the adhesive abilities of PANC-1 and Mia PaCa-2 cells to gelatin-coated wells under different acidic conditions (Figure 3) confirmed that cell-matrix interactions are inhibited under acidic conditions during the early phase of acidotic stress (4 days pH_e_ 6.6), as previously demonstrated in other works [19,29,31]. On the contrary, acid selection enhanced the ability of PDAC cells to adhere, maintaining this feature once PANC-1 cells face back physiological pH_e_ 7.4 (Figure 3a,b). The acidic treatment led to PDAC cells with an increased migratory potential at population (Figure 3c–e) and single-cell level (Appendix A), with further accentuation in cells re-acclimated to pH_e_ 7.4 (Figure 3c–e). Of particular interest are the results obtained from pH_e_-taxis experiments, which confirmed that the same behavior was observed in the presence of a pH_e_ gradient, allowing the cells to migrate from an acidic compartment to a pH_e_ 7.4 one (Figure 4a–c). These data support the hypothesis of acidic selection of a more invasive phenotype as observed on the edges of the tumor. Interestingly, this behavior is not common to immune cells which instead tends to move toward a more acidic pH_e_ [98].

Despite the inhibition of cell attachment of PANC-1 and Mia PaCa-2 cells exposed for 4 days to low pH_e_, those cells migrated faster in acidic conditions. 

The apparent contradictory behavior induced at early stages of acidic pHe selection (4 days), i.e., lower cell-matrix adhesion properties and increased migration velocity, could be explained by an early acidic pH_e_ selection. In line with this hypothesis, the few PANC-1 cells that survived and remained attached to the gelatin-coated surfaces at the early stages of pH_e_ 6.6 exposure might have already evolved sufficiently into a more aggressive phenotype characterized by faster cells, as suggested by the greater percentage of PANC-1 cells characterized by high-velocity respect to control cells (Appendix A). The acquisition of this migratory phenotype might not require longer acidic treatment or acclimation to pH_e_ 7.4. In this context, focal adhesions (FAs) might be involved. They play a key role in cell migration, as these multi-protein assemblies represent the main linkage between the intracellular cytoskeleton and extracellular matrix, promoting membrane protrusion at the leading edge of migrating cells [99]. Our immunostaining results on PANC-1 cells might confirm this, as cells exposed for 4 days to pH_e_ 6.6 were faster than control cells and showed increased paxillin recruitment to the cell periphery, with larger FAs compared to control (Appendix A). On the contrary, the acute acidic treatment did not affect the size of FAs, their number, nor their plasma membrane recruitment in Mia PaCa-2 cells (Appendix A).

One major step in tumor metastasis is local tumor cell invasion, a process promoted by acidic TME [10,21]. Our results indicated that, although PDAC invasive capacities are impaired during the early stages of low pH_e_ selection, they were indeed potentiated during long-term acidic treatment (1 month). More interestingly this pro-metastatic phenotype is conserved following cell re-acclimation to physiological pH_e_ (Figure 3f–h). Transwell invasion experiments performed with pH_e_ gradient (Figure 4d) demonstrated that the increased invasive properties acquired by acid-selected cells were further potentiated when cells found themselves at the interface with less acidic areas mimicking tumor-stroma boundaries, where cells are in contact with better perfused, and therefore less acidic, regions. Indeed, invadopodial activity was increased in PDAC pH_e_-selected + 7.4 cells, leading to a more pronounced digestion of the ECM. Although we did not provide the molecular mechanisms involved, a major role in low pH_e_-induced matrix degradation is expected for Na^+^-H^+^ exchanger (NHE1), localized in the invadopodia [100,101,102] and which promotes the proteolytic activity of different proteinases [46,58,59], and by Carbonic Anhydrase IX (CAIX), which acidifies the extracellular space to promote MMP14 activity [60,103] and metastasis in vivo [104]. 

These results highlight the metastatic potential of pH_e_-selected cells in vitro and indicate that the increased invasiveness results from a gradual selection of aggressive cell phenotypes induced by the acidic pH_e_. However, our results following a short-term acid exposure contrast with previous reports with PDAC cells [20], which may explain the different strategies employed for creating the acidic conditions, and other cancer types [14,25,40,59]. In fact, particular attention should be addressed to the different responses that a specific maneuver of pH_e_ may induce in cancer cells [8]. Our results suggest that the impact of acidic pH_e_ on cell invasion depends on the strategy used for manipulating the pH, which can produce distinct responses based on the deregulation of specific proteins involved in pH_e_/pH_i_ sensing and transduction. Moreover, the adaptation/selection process might produce different outcomes and take different times to activate/inhibit specific cell processes based on the cell type. We found that some extent of selection is observable after 15 days of acid treatment, concurrently with the progressive recovery of cell viability. 

A section of the paper was devoted to underlining the role of acidic pH selection toward a mesenchymal phenotype. Indeed, EMT is another crucial step in disseminating tumor cells from the primary sites to other organs, providing cancer cells with invasive properties [105]. Our work showed that pH_e_ selection induced a mesenchymal phenotype in both PDAC cells, as demonstrated by an increase in the mRNA levels of several EMT markers. Moreover, PANC-1 exposed to pH_e_ 6.6 for 4 days and pH_e_-selected + 7.4 cells showed a significant enrichment of the EMT gene set (as provided by dbEMT 2.0) in our DEG list (Figure 7a,b), including several metalloproteases and integrins genes. Interestingly, among the EMT genes enriched in the pH_e_-selected + 7.4 cells DEG list, we also observed a significant increase in the FGFR2 expression, confirming the role of this receptor in PDAC cell EMT as recently suggested [106]. The EMT phenotype was confirmed by qPCR data, showing a significant decrease in Slug gene expression in the 4 days pH_e_ 6.6 models, which might correlate with the decrease in proliferative and invasive abilities in the early stages of pH_e_ selection, as suggested in colorectal cancer [107]. Several publications support the notion that extracellular acidosis promotes EMT [15,19,26,86]. In our work, RNA sequencing on PANC-1 cells allowed us to identify the different genetic signatures of the short-term and long-term (and recovery to pH_e_ 7.4) acid models, observing the acid-mediated downregulation of genes involved in cell growth, invasion, and adhesion or the upregulation of cell death-related genes during the early stages of pH_e_ selection (Figure 6c and Figure 7c,e,g). We also validated that different gene sets of cell proliferation, ECM remodeling, migration, and invasion were associated with the acid-selected phenotypes recovered to pH_e_ 7.4, whose migratory and invasive phenotypes were related to epithelial–mesenchymal transition induction (Figure 6d and Figure 7d,f,h). These results are in accordance with previous RNA-seq data on 1-month acid-adapted PANC-1 cells [41] and microarray data on other pancreatic cancer cells adapted to low pH_e_ [86]. 

## 5. Conclusions

This work reports that the acidic tumor microenvironment provides certain PDAC cell subpopulations with selective survival benefits that allow them to survive under acidic stress conditions. They do so via genetic reprogramming toward the expression of proliferation, migration, invasion, autophagy, and EMT-related markers, leading to PDAC cells with enhanced metastatic potential which promote cancer progression (Figure 8). 

## Figures and Tables

**Figure 1 cancers-15-02572-f001:**
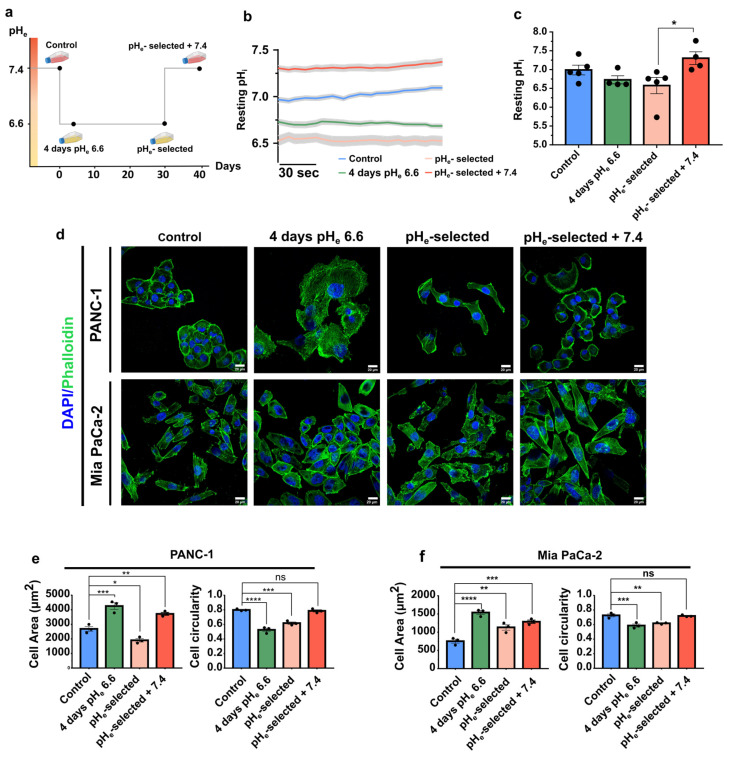
Effects of acidic pH_e_ on PANC-1 cells’ intracellular pH and PDAC cells’ morphology. (**a**) Scheme of the different acidic pH_e_ phenotypes established. Control cells were kept in physiological pH_e_ culture conditions (pH_e_= 7.4), while acidic phenotypes were constituted by PDAC cells exposed for different periods to acidic pH_e_: 4 days (cell model named “4 days pH_e_ 6.6”), 1 month (pH_e_-selected), and 1-month long exposure followed by recovery to physiological pH_e_ for 2 weeks (pH_e_-selected + 7.4). (**b**) Mean traces (± SEM) of at least 4 independent experiments illustrating the resting pH_i_ values in PANC-1 control cells (blue) and the different acidic phenotypes. Cells were loaded with 3 µM BCECF, and pH_i_ values were recorded for 3 min as a fluorescent ratio (490/440 nm) changes following exposition to pH_e_ 6.6 for 4 days (green), 1 month (light orange), and 1-month prior recovery to pH_e_ 7.4 for 2 weeks (red). (**c**) Quantification of pH_i_ values in PANC-1 control and acidic phenotypes from 3 min recording. Each dot indicates the mean value of one independent experiment (*n* ≥ 108 cells for each condition). Data were analyzed using the Kruskal–Wallis *H*-test and Dunn’s multiple comparison test, * *p* < 0.05, ns = not significant. (**d**) Alexa Fluor 488-phalloidin (F-actin, green) and DAPI (nucleus, blue) staining of the different PANC-1 (top) and Mia PaCa-2 (bottom) cell models on a 1% gelatin-coated surface. Scale bar 20 µm. (**e**) Quantification of cell area in µm^2^ (left) and cell circularity index (right) of the different PANC-1 and (**f**) Mia PaCa-2 cell models. Data were reported as mean (± SEM) of 5 representative regions per condition; 3 independent experiments were performed for each condition. Data were analyzed using One-way ANOVA with Dunnett’s multiple comparisons test. * *p* < 0.05, ** *p* < 0.01, *** *p* < 0.001, **** *p* < 0.0001, ns = not significant.

**Figure 2 cancers-15-02572-f002:**
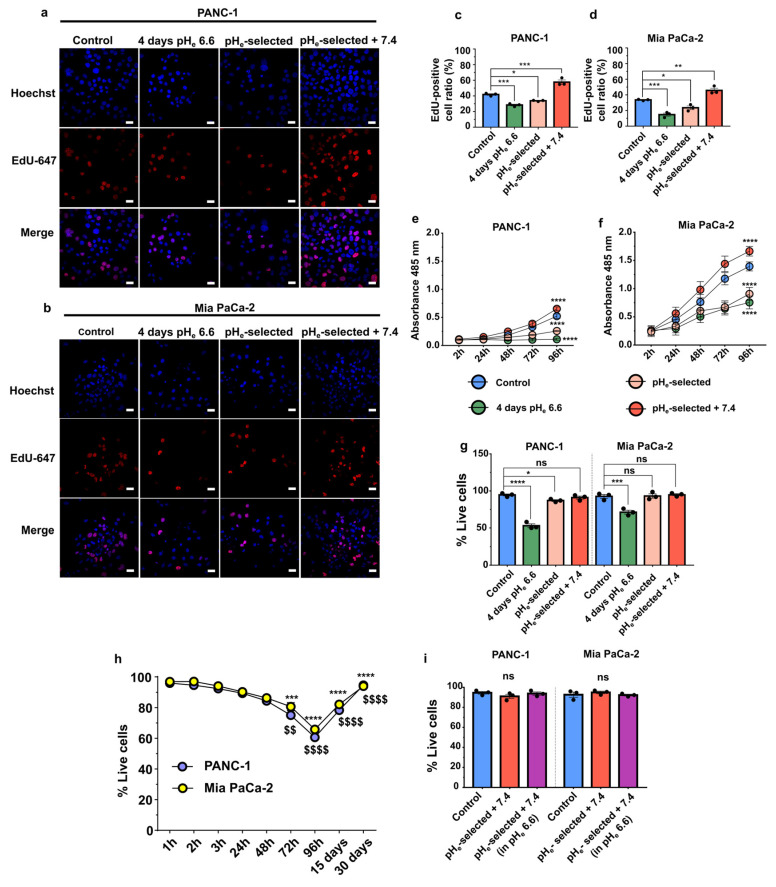
Effects of acidic pH_e_ on PDAC cells’ viability and proliferation. (**a**) Representative fluorescence images of PANC-1 and (**b**) Mia PaCa-2 cell proliferation obtained by EdU staining assay (red, Alexa Fluor 647) and Hoechst (blue) nuclear staining. Scale bar = 20 µm. (**c**) Quantification of the percentage of PANC-1 and (**d**) Mia PaCa-2 EdU-positive cells upon treatment with acidic pH_e_. Data were reported as the percentage of EdU/Hoechst-positive cell mean ± SEM from 4 representative regions for each condition. Time course of proliferation of the different models of (**e**) PANC-1 and (**f**) Mia PaCa-2 cells assessed by an MTS assay. Significant differences between control cells vs. all other conditions at 96 h. (**g**) Determination of PANC-1 (left) and Mia PaCa-2 (right) cell viability by trypan blue exclusion assay in control cells (pH_e_ 7.4), 4 days pH_e_ 6.6, pH_e_-selected (1 month in pH_e_ 6.6) and pH_e_ selected + 7.4 (1 month in pH_e_ 6.6 followed by 2 weeks in pH_e_ 7.4). (**h**) Time course of PANC-1 (blue) and Mia PaCa-2 (yellow) cell viability by trypan blue exclusion assay in control cells exposed to acidic pH_e_ for different times. Significant differences in PANC-1 (*) or Mia PaCa-2 cells ($) at each time point of acidic treatment vs. 1 h. (**i**) Determination of PANC-1 (left) and Mia PaCa-2 (right) cell viability as assessed by trypan blue exclusion assay in control cells (pH_e_ 7.4) and pH_e_ selected + 7.4 (1 month in pH_e_ 6.6 followed by 2 weeks in pH_e_ 7.4) cells in pH_e_ 7.4 and following 96 h treatment in pH_e_ 6.6. Data were presented as mean ± SEM using One-way ANOVA with Dunnett’s multiple comparisons test. All data shown were obtained from three independent experiments, * *p* < 0.05, ** *p* < 0.01, *** *p* < 0.001, **** *p* < 0.0001, $$, *p* < 0.01, $$$$ *p* < 0.0001, ns = not significant.

**Figure 3 cancers-15-02572-f003:**
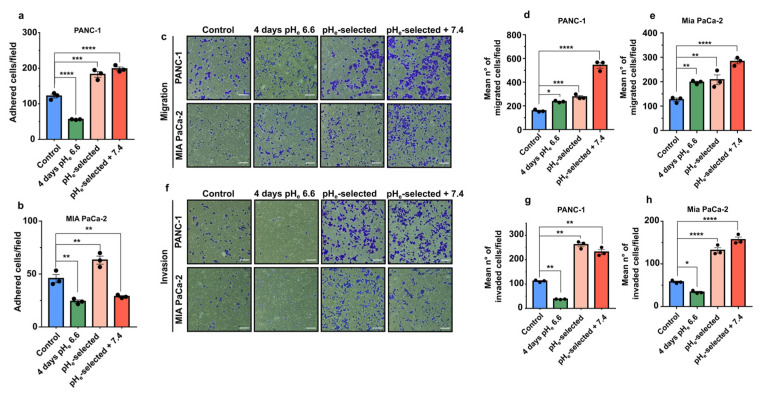
Effects of acidic pH_e_ on PDAC cell adhesion, migration, and invasion. (**a**) Quantification of PANC-1 and (**b**) Mia PaCa-2 cell adhesion assays. Cells were exposed to acidic conditions for different periods. Data were reported as mean ± SEM from at least 4 representative regions for each condition. (**c**) Representative brightfield microscopic images of crystal violet-stained PDAC cell (in blue) that migrated through the transwell membrane in the migration assay. Scale bar = 100 µm. Quantification of the mean number of migrated (**d**) PANC-1 and (**e**) Mia PaCa-2 cells determined in a transwell 18-h long migration assay. Data were reported as mean ± SEM. (**f**) Representative images of PDAC cell models that invaded through the Matrigel-coated transwell membrane in the invasion assay. Scale bar = 100 µm. Quantification of the mean number of invaded cells of the 18-h long transwell invasion assay in (**g**) PANC-1 and (**h**) Mia PaCa-2 cells. Data were reported as mean ± SEM of three independent experiments and analyzed using One-way ANOVA with Dunnett’s multiple comparisons test. All data shown were obtained from three independent experiments, * *p* < 0.05, ** *p* < 0.01, *** *p* < 0.001, **** *p* < 0.0001.

**Figure 4 cancers-15-02572-f004:**
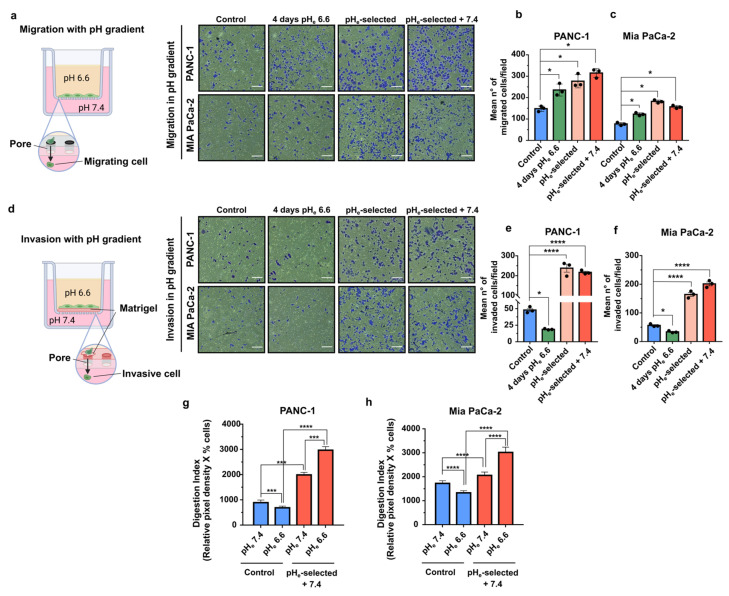
Effects of a pH_e_ gradient on PDAC cells migration, invasion, and invadopodia activity. (**a**) Representative microscopic images of PDAC cells’ migration assay, showing cells that migrated through the transwell membrane in a pH gradient, allowing cells to move from an acidic compartment to the pH 7.4 bottom part of the transwell. Scale bar = 100 µm. (**b**) Quantification of the mean number of migrated cells of the transwell migration assay in PANC-1 and (**c**) Mia PaCa-2 cells in the presence of a pH gradient. (**d**) Representative microscopic images of PDAC cell invasion assay, showing cells that invaded through the Matrigel-coated transwell membrane in a pH gradient. Scale bar = 100 µm. (**e**) Quantification of the mean number of invaded cells of the transwell invasion assay in PANC-1 and (**f**) Mia PaCa-2 cells in the presence of a pH gradient. (**g**) Effect of acidic pH_e_ on the Digestion Index of both PANC-1 and (**h**) Mia PaCa-2 control and pH_e_-selected + 7.4 cells. The percentage of cells that produced invadopodia and their ECM degradation was determined by in situ zymography. The mean total invadopodia proteolytic activity was calculated as follows: Digestion Index = % of cells positive for invadopodia ECM digestion × mean pixel density of focal ECM digestion/cell. Data were reported as mean and ± SEM and analyzed using One-way ANOVA with Dunnett’s multiple comparisons test with Tukey’s multiple comparisons test for (**g**,**h**). All data shown were obtained from three independent experiments, * *p* < 0.05, *** *p* < 0.001, **** *p* < 0.0001.

**Figure 5 cancers-15-02572-f005:**
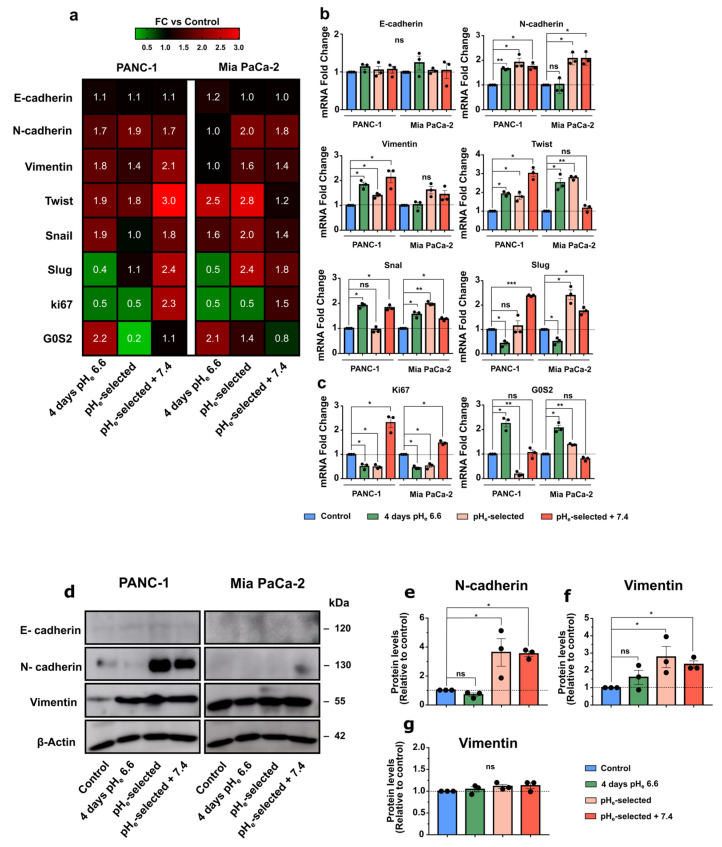
Effects of acidic pH_e_ on epithelial–mesenchymal transition and proliferation markers of PDAC cell. (**a**) Heatmap of mRNA levels for epithelial–mesenchymal transition (EMT) markers in PANC-1 and Mia PaCa-2 cells. Columns represent each condition of the different PDAC cell lines, while rows indicate each differentially expressed gene. The value indicated is the mean value of the fold change of triplicate samples relative to control. Fold changes relative to control are visualized using a green-to-red gradient color scale. (**b**) mRNA expression levels of the different EMT markers and (**c**) proliferation (Ki67) and cell-cycle arrest (G0S2) markers in PANC-1 and Mia PaCa-2 cells subjected to acidic treatment for different periods and presented as fold change values, obtained by RT-qPCR. The effects of pH_e_ 6.6 treatment on EMT and proliferation marker expression were compared with control samples (dotted lines). Fold changes were quantified using the 2^−ΔΔCq^ method and normalized to HPRT reference gene. (**d**) Representative Western blot results that illustrate the effect of different times of exposure to acidic pH_e_ on EMT proteins in PANC-1 and Mia PaCa-2 cells. (**e**) Relative densiometric quantification of Western blot results, showing the abundances of N-cadherin and (**f**) Vimentin proteins in PANC-1 cells and (**g**) Vimentin protein levels in Mia PaCa-2 cells compared to control conditions after normalization with β-actin. Data were presented as mean and ± SEM and analyzed using one-way ANOVA with Dunnett’s multiple comparisons test. All data reported were obtained from three independent experiments, * *p* < 0.05, ** *p* < 0.01, *** *p* < 0.001, ns not significant. The uncropped blots are shown in Appendix A—Uncropped Western Blots Membranes.

**Figure 6 cancers-15-02572-f006:**
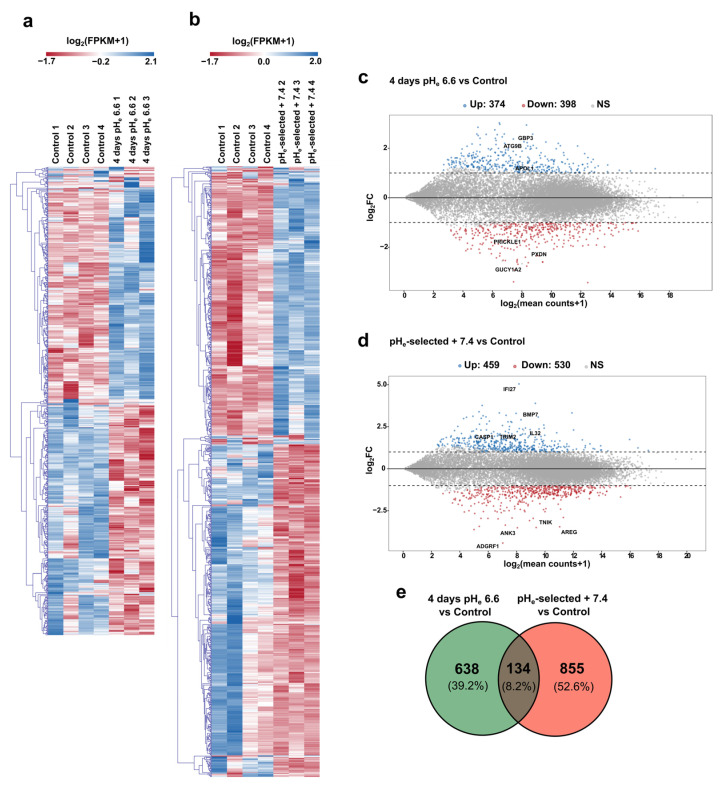
Differential transcriptomic profiles in PANC-1 cells in response to acidosis. (**a**) Heatmaps of RNA-Seq transcriptomic data relative to the differentially expressed genes (DEGs) detected comparing PANC-1 control condition against 4 days pH_e_ 6.6 cells and (**b**) PANC-1 pH_e_-selected + 7.4 cells, respectively. Rows of the heatmaps represent the DEGs detected in the two comparisons, while columns are the individual biological replicates. Gene expression values are shown as gene-wise median-centered “log2(FPKM+1)” (FPKM = Fragments Per Kilobase of sequence per Million mapped reads) according to a red-to-blue color gradient. (**c**) MA-plots showing gene differential expression observed in PANC-1 after 4 days at pH_e_ 6.6 and (**d**) in PANC-1 pH_e_-selected + 7.4 cells compared to Control. “M = log2FC” vs. “A = log2(mean counts+1)”. Downregulated genes are in red, while upregulated DEGs are in blue. Grey dots indicate non-significant gene expression changes. (**e**) Venn diagram summarizing the number of differentially expressed genes in PANC-1 4 days pH_e_ 6.6 cells vs. Control and pH_e_-selected + 7.4 cells vs. Control; the number of genes deregulated in both acidic conditions is represented by the overlap between the two circles.

**Figure 7 cancers-15-02572-f007:**
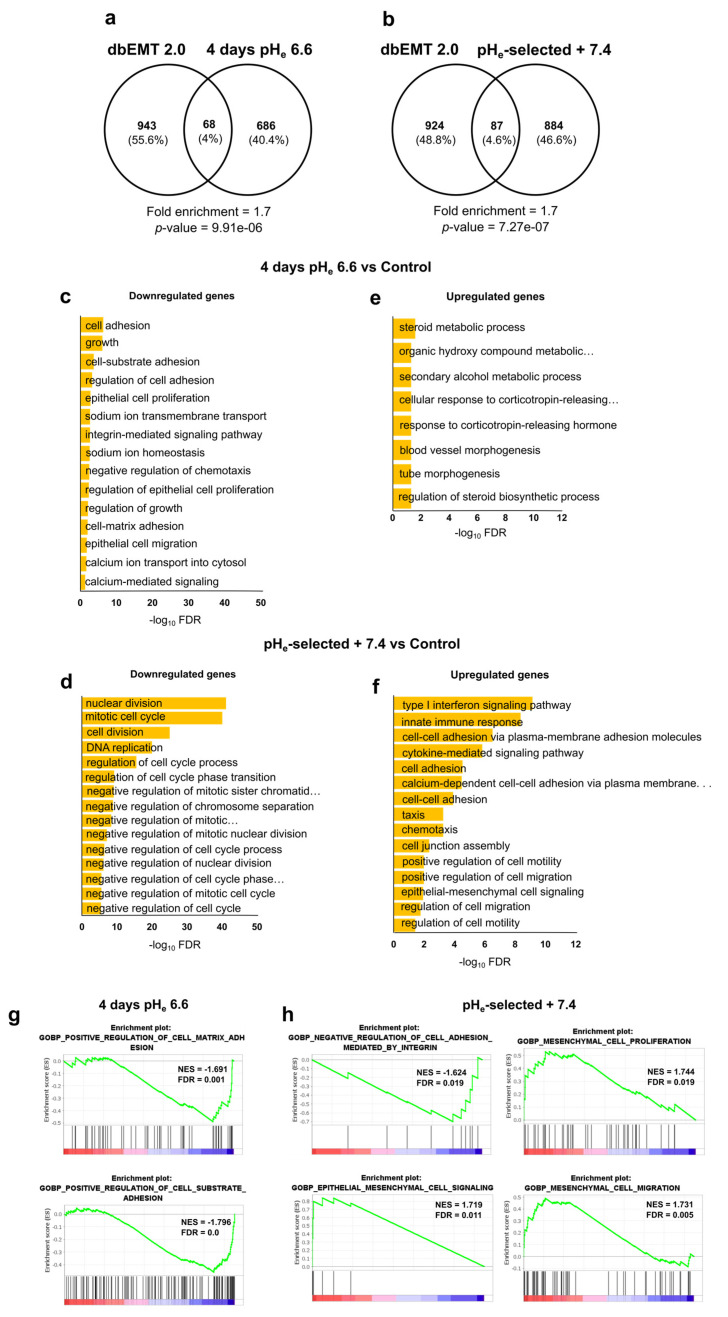
Gene ontology (GO) and gene set enrichment analysis (GSEA) of differentially expressed genes in PANC-1 cells in response to acidosis. (**a**) Venn plots showing overlapped dbEMT2.0 curated genes and differentially expressed genes (DEGs) in PANC-1 4 days pH_e_ 6.6 and (**b**) PANC-1 pH_e_-selected cells + 7.4. (**c**) Bar chart of the most relevant enriched biological processes as defined in GO database and resulting from the analysis of genes downregulated in PANC-1 4 days pH_e_ 6.6 cells vs. Control, and (**d**) in PANC-1 pH_e_-selected + 7.4 cells vs. Control, and of genes upregulated in (**e**) PANC-1 4 days pH_e_ 6.6 cells vs. Control, and (**f**) in PANC-1 pH_e_-selected + 7.4 cells vs. Control. (**g**) RNA-Seq data of PANC-1 4 days pH_e_ 6.6 and (**h**) pH_e_-selected + 7.4 cells analyzed using gene set enrichment analysis (GSEA) and presented as enrichment score (ES) plots. GSEA showed significant enrichment of cell-substrate adhesion, and migration-related gene sets in the two acidic conditions, and of proliferation-related gene set for pH_e_-selected + 7.4. *y*-axes in GSEA plots represent the ES function, while *x*-axes display a red-blue color scale corresponding to the ranked list of DEGs (from the most up- to the most downregulated one, respectively) following one of the two acidic treatments. Vertical black lines above the red-blue scale in each plot refer to the position of each gene of the selected gene set along the ranked gene list as returned by the differential expression analysis of experimental data.

**Figure 8 cancers-15-02572-f008:**
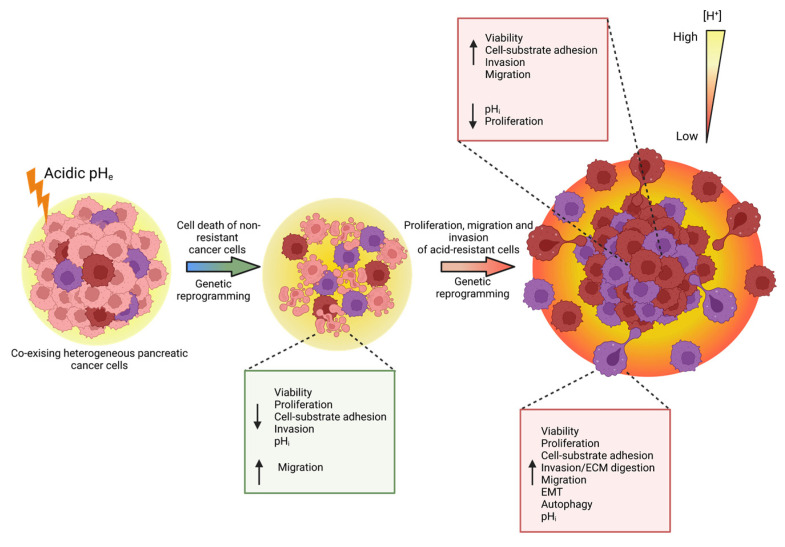
PDAC cells exposed to acidic extracellular conditions undergo a process of selection, characterized by acid-induced genetic and phenotypic alterations. This results in increased cell death due to the cytotoxic effect of low pH_e_ and decrease in cell-substrate adhesion, proliferation, and cell invasion. Along with the acid exposition, further genetic rewiring provides surviving cancer cells with more aggressive properties in terms of adhesion, migration, and invasion. Limited proliferative capacities are overcome when cells are acclimated to pH_e_ 7.4 following 1 month-long pH_e_ 6.6 treatment. **↓** arrow indicates downregulation, ↑ arrow indicates upregulation. The figure was created with www.Biorender.com.

**Table 1 cancers-15-02572-t001:** List of primers for qPCR.

Gene Name	Primer Probe	Sequence (5′ to 3′)
hE-cadherin	Forward	GAACGCATTGCCACATACAC
	Reverse	GAATTCGGGCTTGTTGTCAT
hN-cadherin	Forward	CCTGAGGGATCAAAGCCTGGAAC
	Reverse	TTGGAGCCTGAGACACGATTCTG
hVimentin	Forward	TCTACGAGGAGGAGATGCGC
	Reverse	GGTCAAGACGTGCCAGAGAC
hSnaiI	Forward	CTTCCAGCAGCCCTACGAC
	Reverse	CGGTGGGGTTGAGGATCT
hTwist	Forward	AGCAAGATTCAGACCCTCAAGCT
	Reverse	CCTGGTAGAGGAAGTCGATGTACCT
hSlug	Forward	TGTTTGCAAGATCTGCGGC
	Reverse	TGCAGTCAGGGCAAGAAAAA
hKi67	Forward	TGACCCTGATGAGAAAGCTCAA
	Reverse	CCCTGAGCAACACTGTCTTTT
hG0S2	Forward	AAGGGGAAGATGGTGAAGCTG
	Reverse	CTGCACACAGTCTCCATCAGG
hHPRT1	Forward	AGTTCTGTGGCCATCTGCTT
	Reverse	CAATCCGCCCAAAGGGAACT

## Data Availability

The data presented in this study are available in this article and Appendix A.

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
