# Peer review of "Acidic Growth Conditions Promote Epithelial-to-Mesenchymal Transition to Select More Aggressive PDAC Cell Phenotypes In Vitro"

_cancers, 2023, doi:10.3390/cancers15092572_

Round 1
Reviewer 1 Report
The manuscript by Audero et al explores the effects of low pH culturing conditions on two PDAC cell lines in vitro on selection of more aggressive cell phenotypes. They describe the effects on cell proliferation, migration and invasion in vitro as well as on expression of several EMT markers.
Given the limitation of the work to the in vitro conditions and inclusion of one variable of the environment I would first suggest the revision of the title to more appropriately reflect the extent of the work that has been done to:
Acidic growth conditions (instead of the microenvironment) promote ….. to select for more aggressive PDAC cell phenotypes in vitro
As no microenvironmental components of the stroma/fibroblasts, stellate cells, matrix or immune cell types were included in the experimental design, I think the current title is too generalized for the work that has been done.
There are also minor inconsistencies and overstatements throughout the manuscript, as well as some more prominent omissions, here listed in order they appear in the text:
1. Line 53 “far below” instead of “far beyond”
2. Line 382, 383 states that the effects are not significant
difference is not statistically significant (pHi 6.99 ± 0.12 control conditions; pHi 6.72 ± 0.10 4 days pHe 6.6; pHi 6.57 ± 0.21 pHe-selected cells)
And then the conclusion states that the results are significant
Line 387, 388
results indicate that extracellular acidosis leads to intracellular acidifica- tion, significantly affecting PANC-1 cells exposed to acidic pHe for extended periods
3. In the Figure 1D and F for MiaPaca2 quantification, the selected image at pH6.6 4 days does not show three times bigger cells, as it is stated in the quantification graph, where the size supposedly increased from 500 to 1500uM2.
4. The major methodological question arises as to why was the coating with gelatin chosen in some but not all experiments, as both PANC1 and MiaPaca2 adhere to plastic an glass without problems? There are two obstacles in interpreting results on gelating coating in the light of adhesion/migration:
1. MiaPaca2 cell line does not express collagen binding integrins α1 and α2, and seeding on gelatin changes its adhesion behavior, making it less attached to the surface
2. More importantly, pH influences gelatin crosslinking and viscosity and imposes new parameter that can influence cell adhesion through integrin signaling in these experiments for both cell lines.
5. Methods section states MTS assay results section refers to MTT assay
6. In MTS assay cells seeded on plastic, in BrdU assay seeded on gelatin. In trypan blue again on plastic.
7. MTT/MTS reduction is also pH dependent. The rate of tetrazolium reduction reflects the general metabolic activity or the rate of glycolytic NADH production and can change with culture conditions pH and glucose content of medium. Please restate the results of the assay accordingly, as they are not the measure of proliferation.
8. Please show the scale for viability in Figure 2H from 0 to 100% as in other graphs in the figure.
9. The results decsribed in the section on proliferation in acidic conditions are already known, as it is well established that proliferation of mammalian cells is dependent on a permissive pHi in the slightly alkaline range (7.0-7.2)
and that mitogen signaling associated with an intracellular alkalinization.
10. Supplementary figure 1C and D please put the same scale 250 cells/field so that it is clear that MiaPaca2 adhere more slowly.
11. Please set the scale in Figure 2E and 2F to the same scale.
Seeded twice as many MiaPaca2 cells an they adhere half of the numbers compared to PANC1.
12. Title in line 423 Extracellular acidification inhibits proliferation change to decreases proliferation
13. Line 432 change to pH7.4 allows recovery and not increase proliferation
14. Please explain in the Results section the decreased numbers of adherent cells, yet more cells per field in Figure 1D nad Figure 2A and B. Authors refer to this somewhat in the discussion, but the interpretation here is lacking.
15. Please revise the sentence in line 463 “These results indicate that, although acute acidification promotes significant cell death in both PDAC cell lines, the reduced proliferation of pHe-selected PDAC cells is not due to cell death.” From the data it is evident that acidification induces cell death and selects for a subpopulation of cells that outgrows and is viable under acidic conditions, and even has higher proliferation rate, which is consistent with more aggressive phenotype.
16. Is the increased adhesion in selected population dependent on integrin expression? Please provide the data on FA assembly for the MiaPaca2 cell line too.
17. What it the timeframe for migration and invasion experiments?
18. What is the explanation for the PANC1 cell line in migration experiments where 50% of cells are dead after 4 days in acidic conditions, and there are more migrated cells than in control?
19. Please rephrase in the text describing Twist, Slug, Snail that it refers to the mRNA, since the protein levels were not presented.
20. Please in Figure 5D show the WB data for the protein levels of vimentin, N cadherin etc in the MiaPaca2 cell line.
21. Please show short tables with individual MMP, EMT genes and integrin expression from the RNA Seq od the PANC1 cell line, to corroborate the GSEA results, as NES scores in the graphs are low.
Lastly, but most importantly, it has already been shown in several cancer cell models that exposure to acidic conditions induces metabolic rewiring, extracellular matrix remodeling and altered cell cycle regulation; as well as that extracellular acidosis promotes EMT.
Authors did cite the papers where the similar work has been done:
“Cancer Cell Acid Adaptation Gene Expression Response Is Correlated to Tumor-Specific Tissue Expression Profiles and Patient Survival” published in Cancers in 2020 by Yao et al. where sequencing of the PANC1 cell line adapted to a low pH sequenced has been published.
And “Identification of Distinct Slow Mode of Reversible Adaptation of Pancreatic Ductal Adenocarcinoma to the Prolonged Acidic PH Microenvironment. By Wu el al J Exp Clin Cancer Res 2022
It is therefore Editor’s discretion to determine whether this manuscript is novel enough for publication in Cancers.
Author Response
The manuscript by Audero et al explores the effects of low pH culturing conditions on two PDAC cell lines in vitro on selection of more aggressive cell phenotypes. They describe the effects on cell proliferation, migration and invasion in vitro as well as on expression of several EMT markers.
The authors thank the Reviewer for the time and effort in reviewing the manuscript and the constructive criticism which allowed us to improve the quality of our work.
Given the limitation of the work to the in vitro conditions and inclusion of one variable of the environment I would first suggest the revision of the title to more appropriately reflect the extent of the work that has been done to:
- Acidic growth conditions (instead of the microenvironment) promote ….. to select for more aggressive PDAC cell phenotypes in vitro.
As no microenvironmental components of the stroma/fibroblasts, stellate cells, matrix or immune cell types were included in the experimental design, I think the current title is too generalized for the work that has been done.
We thank the Reviewer for this suggestion; the title of the work has been modified accordingly.
There are also minor inconsistencies and overstatements throughout the manuscript, as well as some more prominent omissions, here listed in order they appear in the text:
- Line 53 “far below” instead of “far beyond”
We thank the Reviewer for this correction. We have corrected it accordingly.
- Line 382, 383 states that the effects are not significant difference is not statistically significant (pHi 6.99 ± 0.12 control conditions; pHi 6.72 ± 0.10 4 days pHe 6.6; pHi 6.57 ± 0.21 pHe-selected cells). And then the conclusion states that the results are significant
Line 387, 388 results indicate that extracellular acidosis leads to intracellular acidification, significantly affecting PANC-1 cells exposed to acidic pHe for extended periods
We thank the Reviewer for this observation, and we have modified the text by removing “significantly” in line 391.
- In the Figure 1D and F for MiaPaca2 quantification, the selected image at pH 6.6 4 days does not show three times bigger cells, as it is stated in the quantification graph, where the size supposedly increased from 500 to 1500uM2.
We agree with the Reviewer that the chosen image for Mia PaCa-2 cells exposed to pHe 6.6 for 4 days might not be the best representative. Our quantification demonstrated that these cells have a double cell area (1539 µm2 ± 55.86) compared to control cells (753 µm2 ± 55.95). We have replaced the panel in question in Figure 1D with a figure better representing this difference.
- The major methodological question arises as to why was the coating with gelatin chosen in some but not all experiments, as both PANC1 and MiaPaca2 adhere to plastic an glass without problems? There are two obstacles in interpreting results on gelating coating in the light of adhesion/migration:
- MiaPaca2 cell line does not express collagen binding integrins α1 and α2, and seeding on gelatin changes its adhesion behavior, making it less attached to the surface
- More importantly, pH influences gelatin crosslinking and viscosity and imposes new parameter that can influence cell adhesion through integrin signaling in these experiments for both cell lines.
We thank the reviewer for rising this point. We are aware that acidic pH can influence gelatin crosslinking and viscosity. However, it has to be noticed that in our experiments the crosslinking of gelatine coating was performed at physiological pH before seeding, and therefore we can exclude any effect of acidity in this process.
However, to better clarify the adhesion properties of PDAC cells, we performed new experiments on both cell lines in the absence of gelatine as substrate. As shown in the following figure, we confirmed the cell adhesion behaviour of both PANC-1 and Mia PaCa-2 cell lines even in the absence of gelatine as substrate (compare with Figure 3a, b in the main text).
We did not include this data in the text but just for the Reviewer's rebuttal (Fig 1 of the present letter).
Figure 1Quantification of PANC-1 (left) and Mia PaCa-2 (right) cell adhesion assays in absence of gelatin coating. Data were reported as mean ± SEM from 3 independent experiments.
- Methods section states MTS assay, results section refers to MTT assay
We thank the Reviewer for this comment. We have corrected lines 433 and 437 with the correct assay name.
- In MTS assay cells seeded on plastic, in BrdU assay seeded on gelatin. In trypan blue again on plastic.
We thank the reviewer for this observation. To be more consistent, we replaced the EdU assay performed on gelatine (previous Figure 2a - d, now Supplementary Figure 1a, b) with the new Figure 2 a - d performed with cells plated on plastic. The method section has been modified accordingly. The results obtained with cells seeded on plastic completely confirmed our previous results performed with gelatine as cell substrate (see quantification below, Fig 2 of the present letter).
Figure 2 Quantification of the percentage of PANC-1 (left) and Mia PaCa-2 (right) EdU-positive cells upon treatment with acidic pHe. Data were reported as the percentage of EdU/Hoechst-positive cell mean ± SEM from 3 independent experiments.
- MTT/MTS reduction is also pH dependent. The rate of tetrazolium reduction reflects the general metabolic activity or the rate of glycolytic NADH production and can change with culture conditions pH and glucose content of medium. Please restate the results of the assay accordingly, as they are not the measure of proliferation.
We thank the reviewer for the comment. We perfectly agree with the Reviewer, and we are aware that MTT/MTS reduction is pH dependent. Indeed, as already clarified in the text (lines 428-430), we supported the MTS results by quantifying the percentage of DNA-synthesizing cells (Figure 2a - d) and the ATP production (Supplementary Figure 1c, d). MTS/MTT assay is commonly used to measure the proliferative state of cells in culture. Indeed, the quantity of formazan product as measured by absorbance at 490nm is directly proportional to the number of living cells in culture (see also Cory, A.H. et al. (1991) Cancer Commun, DOI: 10.3727/095535491820873191; Riss, T.L. and Moravec, R.A. (1992). Methods Mol Biol, DOI: 10.1007/978-1-61779-108-6_12).
We believe our data reveal the role of the different extracellular acidification treatments on cell proliferation since the data were confirmed by 3 independent methods (EdU incorporation (Figures 2a - d); MTS (Figures 2e, f) and ATP measurements (Supplementary Figures 1c, d).
- Please show the scale for viability in Figure 2H from 0 to 100% as in other graphs in the figure.
We thank the Reviewer for this suggestion. We have modified Figure 2H with the new scale from 0 to 100%.
- The results described in the section on proliferation in acidic conditions are already known, as it is well established that proliferation of mammalian cells is dependent on a permissive pHi in the slightly alkaline range (7.0-7.2) and that mitogen signaling associated with an intracellular alkalinization.
We thank the Reviewer for this clarification. Although the anti-proliferative role of acidic extracellular pH is well established in the literature in different cancers, also in PDAC, there is a lack of evidence of the effect of acidic selection and recovery to physiological pH on cell proliferation, which represents one of the novel aspects of our manuscript. To our knowledge, only two recent publications (J. Schnipper et al., 2022, doi:10.3390/cancers14194946; Czaplinska et al, 2022, https://doi.org/10.1002/ijc.34367) have demonstrated that acid adaptation and recovery to pH 7.4 restore proliferation rates in PANC-1 cells. However, differently from Schnipper J. et al., our results demonstrate that the recovery in pH 7.4 not only restores but further enhances the cell proliferation in both PDAC cell lines as compared to control conditions (see also Figure 2 of the results section). On the other hand, the results presented by Czaplinska D. et al. seem to point out an enhancement in the spheroid area when recovered at pH 7.4, which is not always significant depending on the cell substrate. It has to be noted that Czaplinska and coauthors did not adopt a selection model (as described in our paper) but rather an adaptation model to acidic conditions (by lowering the pH gradually during the adaptation).
- Supplementary figure 1C and D please put the same scale 250 cells/field so that it is clear that MiaPaca2 adhere more slowly.
We thank the Reviewer for this suggestion. We have modified Supplementary Figure 1d (now Supplementary Figure 1e) with the same scale as 1c (now Supplementary Figure 1f).
- Please set the scale in Figure 2E and 2F to the same scale. Seeded twice as many MiaPaca2 cells an they adhere half of the numbers compared to PANC1.
We thank the Reviewer for this suggestion. We have modified Figure 2e with the same scale as Figure 2f. It has to be noticed that a direct comparison between the two cell lines is not appropriate since we did not adopt the exact same conditions for plating (3000 cells/well for PANC-1 and 8000 cells/well for Mia PaCa-2, line 147 of the Methods section).
- Title in line 423 Extracellular acidification inhibits proliferation change to decreases proliferation
We thank the Reviewer for the suggestion, and we proceed to change the verb in question.
- Line 432 change to pH7.4 allows recovery and not increase proliferation
We are not sure we understood the comment. We agree that pHe 7.4 re-exposure after acidic selection allows the recovery of cell proliferation. However, we clearly demonstrated that there is a significant further increase in proliferation as compared with both acidic selection (pH-selected) as well as control conditions (cells grown in pH 7.4, control) as reported in Fig. 2c - d and Fig 2 e - f. To be clearer, we emphasize this aspect in the results section (see lines 438-440).
We believe that this result is quite important since the description of a more aggressive phenotype acquired by PDAC cell lines after acidic selection and recovery in pH 7.4 represents a novel aspect of the manuscript.
- Please explain in the Results section the decreased numbers of adherent cells, yet more cells per field in Figure 1D and Figure 2A and B. Authors refer to this somewhat in the discussion, but the interpretation here is lacking.
We thank the Reviewer for this suggestion. The cell images of Figures 1D and 2A and B (now Supplementary Figures 1a, b) were chosen as representatives for each pH condition. However, we admit that the choices were not the most appropriate to represent the adhesive capacities of the different acidic cell models. We have therefore changed the aforementioned figures with better representative images.
- Please revise the sentence in line 463 “These results indicate that, although acute acidification promotes significant cell death in both PDAC cell lines, the reduced proliferation of pHe-selected PDAC cells is not due to cell death.” From the data it is evident that acidification induces cell death and selects for a subpopulation of cells that outgrows and is viable under acidic conditions, and even has higher proliferation rate, which is consistent with more aggressive phenotype.
We thank the Reviewer for this suggestion. We have rephrased between lines 475 and 478 as follow:
“These results indicate that acute acidification promotes significant cell death in both PDAC cell lines (Figure 2h) within the first 96h after seeding, selecting a subpopulation of cells that outgrows and is viable under acidic conditions, with a higher proliferation rate as compared to non-selected PANC-1 cells (Figure 2e,* and $ symbol), which is consistent with more aggressive phenotype“.
- Is the increased adhesion in selected population dependent on integrin expression? Please provide the data on FA assembly for the MiaPaca2 cell line too.
We thank the Reviewer for the suggestion. We have performed paxillin immunostaining on Mia PaCa-2 cells, and the results were included in Supplementary Figure 2c - i and the text has been modified accordingly. Our results showed that acute acidic treatment (4 days) did not alter the size, the amount nor the periphery recruitment of FAs.
- What it the timeframe for migration and invasion experiments?
The timeframe for migration and invasion experiments in absence of a pH gradient was 18 hours (overnight), and it was added in the corresponding section under Materials and Methods (line 207). Moreover, the timeframe was further indicated in the caption of Figure 3 (lines 500 and 502) to make it clearer to the readers. On the contrary, the timeframe for transwell assays with pH gradient (5 hours) was already specified in the Materials and Methods section.
- What is the explanation for the PANC1 cell line in migration experiments where 50% of cells are dead after 4 days in acidic conditions, and there are more migrated cells than in control?
We clarified and provided a possible explanation for the apparently contradictory behavior in the discussion section (lane 828-842):
“Despite the inhibition of cell attachment of PANC-1 and Mia PaCa-2 cells exposed for 4 days to low pHe, those cells migrated faster in acidic conditions. The apparent contradictory behavior induced at early stages of acidic pHe selection (4 days), i.e., lower cell-matrix adhesion properties and increased migration velocity, could be explained by an early acidic pHe selection. In line with this hypothesis, the few PANC-1 cells that survived and remained attached to the gelatin-coated surfaces at the early stages of pHe 6.6 exposure might have already evolved sufficiently into a more malignant phenotype characterized by faster cells, as suggested by the greater percentage of PANC-1 cells characterized by high-velocity respect to control cells (Supplementary Figure 2d). The acquisition of this migratory phenotype might not require longer acidic treatment or acclimation to pHe 7.4. In this context, focal adhesions (FAs) might be involved. They play a key role in cell migration, as these multi-protein assemblies represent the main linkage between the intracellular cytoskeleton and extracellular matrix, promoting membrane protrusion at the leading edge of migrating cells [99]. Our immunostaining results on PANC-1 cells might confirm this, as cells exposed for 4 days to pHe 6.6 were faster than control cells and showed increased paxillin recruitment to the cell periphery, with larger FAs compared to control (Supplementary Figures 2 c - f)”.
- Please rephrase in the text describing Twist, Slug, Snail that it refers to the mRNA, since the protein levels were not presented.
We thank the Reviewer for the suggestion. We have modified the text in lines 618, 623, and 624.
- Please in Figure 5D show the WB data for the protein levels of vimentin, N cadherin etc in the MiaPaca2 cell line.
We thank the Reviewer for the comment. We have performed Western Blot to assess the protein expression of E-cadherin, N-cadherin, and Vimentin in Mia PaCa-2 cells. The results were included in Figure 5, and the text and figure legend have been changed accordingly. Representative cropped blots for the expression of the aforementioned markers in Mia PaCa-2 were included in Figure 5d, while the quantification of Vimentin was added in Figure 5g. The uncropped membranes have been included in the File Western Blots membranes. The results confirm qPCR data, with Mia PaCa-2 cells expressing very low and undetectable levels of E-cadherin and N-cadherin, independently of acidic treatment (this is why we did not quantify the protein levels). Moreover, Vimentin protein levels were not significantly affected by acid exposure.
- Please show short tables with individual MMP, EMT genes and integrin expression from the RNA Seq od the PANC1 cell line, to corroborate the GSEA results, as NES scores in the graphs are low.
We thank the reviewer for the suggestion. Fig.7a, b show the degree of enrichment of the EMT gene set in our DEG list in pHe-selected + 7.4 cells as well as in 4 days pHe 6.6 cells. To further corroborate the role of EMT in acidosis selection of PANC-1, we included two new supplementary tables listing all DEGs involved in EMT as provided by dbEMT 2.0 (http://dbemt.bioinfo-minzhao.org). Indeed the list shows respectively 87 and 68 significantly deregulated genes (new Supplementary files S3 and S4.), representing the overlap between our DEG list and the set of genes involved in EMT as provided by dbEMT 2.0 (Fig 7a, b). Moreover, as requested by the reviewer, among the EMT-related genes, we highlighted MMP and Integrins genes in the list.
Lastly, but most importantly, it has already been shown in several cancer cell models that exposure to acidic conditions induces metabolic rewiring, extracellular matrix remodeling and altered cell cycle regulation; as well as that extracellular acidosis promotes EMT.
Authors did cite the papers where the similar work has been done:
“Cancer Cell Acid Adaptation Gene Expression Response Is Correlated to Tumor-Specific Tissue Expression Profiles and Patient Survival” published in Cancers in 2020 by Yao et al. where sequencing of the PANC1 cell line adapted to a low pH sequenced has been published.
And “Identification of Distinct Slow Mode of Reversible Adaptation of Pancreatic Ductal Adenocarcinoma to the Prolonged Acidic PH Microenvironment. By Wu el al J Exp Clin Cancer Res 2022
It is therefore Editor’s discretion to determine whether this manuscript is novel enough for publication in Cancers.
We thank the reviewer for giving us the opportunity to better clarify this point which is indeed critical. In the present paper, we assessed the impact of acidosis on PDAC metastatic potential in vitro. We agree with the reviewer that similar models were previously adopted and described. However, a novel aspect of our work is the aggressive behavior presented by the PDAC cells selected in acidic pH when they face back the physiological pHe 7.4. This experimental condition would mimic the interface between acidic cancer edges and neighboring healthy tissues and blood vessels. Indeed, none of the previously published papers shows the role of acidic selection on local invasion toward physiological pH, as well as EMT phenotype acquisition in PDAC. The paper by Wu el al. (10.1186/s13046-022-02329-x) does not take into account the PDAC cell model of recovery after acidic selection; On the other hand, the paper by Yao et al. (10.3390/cancers12082183), as well as Schnipper et al.(10.3390/cancers14194946) use the model of PDAC cells acid-adapted and then recovered back in pH 7.4, but these works do not study the EMT and invasiveness properties as we performed in the present paper in terms of cell invasion behavior, EMT genes deregulation. Therefore, our results support the hypothesis of acidic selection of a more invasive phenotype as observed on the edges of the tumor adding important knowledge on the aggressive behavior observed in PDAC cells facing acidic stress, a key chemical signature of PDAC microenvironment.

Reviewer 2 Report
Authors in this manuscript studied the role of acidic tumor microenvironment in promoting the EMT transition to select more aggressive PDAC cell phenotype. This is a well performed study and I only few minor comments to address which are given below:
1. Reference numbers in the text – lacking space at some places.
2. Rephrase this sentence- After 15 days-long acidic treatment, the percentage of live cells reaches recovers, reaching almost 462 90% after 1 month in low pHe conditions (Figure 2h).
3. Scale bars are missing for the microscopic images in Figure 3- c, f and Figure 4 a, d
4. Figure 5 d- Vimentin levels almost look same. But the quantification shows, vimentin levels are increased in pH- selected and pH-selected +7.4 conditions. Authors should address this issue. They can re-do the blot and include better representative image.
5. Figure 7- a, b: Format these figures. Take out the rectangular box highlighting the figures.
Author Response
Authors in this manuscript studied the role of acidic tumor microenvironment in promoting the EMT transition to select more aggressive PDAC cell phenotype. This is a well performed study and I only few minor comments to address which are given below:
The authors sincerely thank the Reviewer for the insightful comments and recommendations to ameliorate our work.
- Reference numbers in the text – lacking space at some places.
We thank the Reviewer for the observation. The entire text was rechecked, and the spaces in the reference number were fixed.
- Rephrase this sentence- After 15 days-long acidic treatment, the percentage of live cells reaches recovers, reaching almost 90% after 1 month in low pHe conditions (Figure 2h).
We thank the Reviewer for this remark. The mistake was corrected and, in addition, lines 472-476 have been rephrased to make it clearer to the readers.
- Scale bars are missing for the microscopic images in Figure 3- c, f and Figure 4 a, d
We thank the Reviewer for this observation, the scale bars were added in Figures 3c,f and 4a,d, and the legend was modified accordingly.
- Figure 5 d- Vimentin levels almost look same. But the quantification shows, vimentin levels are increased in pH- selected and pH-selected +7.4 conditions. Authors should address this issue. They can re-do the blot and include better representative image.
We thank the Reviewer for this suggestion, and we agree that the chosen blot was not representative of the increased Vimentin protein levels in the PANC-1 cells exposed to long acidic treatment. To address this point, we have substituted the Vimentin panel in Figure 5d with another cell passage that better shows Vimentin overexpression in the aforementioned conditions. The uncropped membrane has been included in the File Western Blots membranes.
- Figure 7- a, b: Format these figures. Take out the rectangular box highlighting the figures.
We thank the Reviewer for this remark. Figures 7a and b have been corrected.
We believe the manuscript has been strongly improved and could be now accepted for publication
Sincerely,
Alessandra Fiorio Pla